# TransLeish: Identification of membrane transporters essential for survival of intracellular *Leishmania* parasites in a systematic gene deletion screen

Andreia Albuquerque-Wendt[1,2,3,9], Ciaran McCoy[2,10], Rachel Neish[4], Ulrich Dobramysl[5], Çağla Alagöz[6,7], Tom Beneke[2,11], Sally A. Cowley[8], Kathryn Crouch[1], Richard J. Wheeler[5,12], Jeremy C. Mottram[4] & Eva Gluenz[1,2,9] ✉

For the protozoan parasite *Leishmania*, completion of its life cycle requires sequential adaptation of cellular physiology and nutrient scavenging mechanisms to the different environments of a sand fly alimentary tract and the acidic mammalian host cell phagolysosome. Transmembrane transporters are the gatekeepers of intracellular environments, controlling the flux of solutes and ions across membranes. To discover which transporters are vital for survival as intracellular amastigote forms, we carried out a systematic loss-of-function screen of the *L. mexicana* transportome. A total of 312 protein components of small molecule carriers, ion channels and pumps were identified and targeted in a CRISPR-Cas9 gene deletion screen in the promastigote form, yielding 188 viable null mutants. Forty transporter deletions caused significant loss of fitness in macrophage and mouse infections. A striking example is the Vacuolar H$^+$ ATPase (V-ATPase), which, unexpectedly, was dispensable for promastigote growth in vitro but essential for survival of the disease-causing amastigotes.

Transmembrane transporters, pumps and channels facilitate the passage of solutes that are otherwise impermeant to lipid bilayers, including sugars, phospholipids, amino acids, ions, drugs and toxins, which need to be shuttled into and out of the cell and distributed across cellular organelles. These transport processes are fundamental to cellular physiology and the maintenance of homoeostasis. For intracellular pathogens, they are at the interface between host and microbe, enabling parasitic scavenging of nutrients[1,2] and supporting

[1]School of Infection and Immunity, University of Glasgow, Sir Graeme Davies Building, 120 University Place, Glasgow G12 8TA, UK. [2]University of Oxford, Sir William Dunn School of Pathology, South Parks Road, Oxford OX1 3RE, UK. [3]Global Health and Tropical Medicine, Instituto de Higiene e Medicina Tropical, Universidade Nova de Lisboa, Rua da Junqueira 100, 1349-008 Lisbon, Portugal. [4]York Biomedical Research Institute, Department of Biology, University of York, York YO10 5DD, UK. [5]Medawar Building for Pathogen Research, Nuffield Department of Medicine, University of Oxford, Oxford, UK. [6]Institute of Cell Biology, University of Bern, Baltzerstrasse 4, 3012 Bern, Switzerland. [7]Graduate School for Cellular and Biomedical Sciences, University of Bern, Bern, Switzerland. [8]James and Lillian Martin Centre for Stem Cell Research, Sir William Dunn School of Pathology, University of Oxford, Oxford OX1 3RE, UK. [9]Present address: Institute of Cell Biology, University of Bern, Baltzerstrasse 4, 3012 Bern, Switzerland. [10]Present address: Animal Physiology and Neurobiology, KU Leuven, 3000 Leuven, Belgium. [11]Present address: Department of Cell and Developmental Biology, Biocentre, University of Würzburg, Am Hubland, 97074 Würzburg, Germany. [12]Present address: Institute of Immunology and Infection Research, School of Biological Sciences, University of Edinburgh, Ashworth Laboratories, Charlotte Auerbach Road, Edinburgh EH9 3FL, UK. ✉e-mail: eva.gluenz@unibe.ch

pathogen cellular physiology in their intracellular niche[3,4]. Protists of the genus *Leishmania* (Order Kinetoplastida) are parasites with a dixenous life cycle, shuttling between an insect vector and a vertebrate host. During its life cycle, the single-celled parasite assumes morphologically and physiologically distinct forms that enable it to colonise the alimentary tract of blood-feeding female phlebotomine sand flies when taken up in a blood meal and persist in the midgut. The parasites then replicate as extracellular flagellated forms, which migrate to the stomodeal valve and differentiate to metacyclic promastigotes preadapted to initiate the infection of a vertebrate host[5,6]. On deposition in the dermis of a mammal during the blood meal, the metacyclic promastigotes are engulfed by resident phagocytic cells. Inside the maturing phagolysosome, the parasites differentiate to the acid-tolerant amastigote form[7], which persist in the resulting parasitophorous vacuole (PV) where they slowly replicate[8] until ingested by another sand fly.

Inside the host cell, principally macrophages, parasite membrane transporters are crucial for meeting two main challenges: Firstly, the parasite must maintain its cellular integrity in an acidic milieu and antimicrobial activities from the macrophage which exposes the parasite to hydrolases, reactive oxygen and nitrogen species. Whilst residing in an environment of ~pH 4-5, amastigotes need a set of surface membrane transporters adapted to functioning at a low external pH. At the same time, they must manage the flow of protons across their membranes to maintain a cytosolic pH of ~pH 6.5[9,10] and a membrane potential of -100 mV[11] involving the activity of proton pumps that use energy from ATP to build up proton gradients across membranes. A cell-surface localised P-type $H^+$ ATPase is thought to be the principal regulator of amastigote intracellular pH[12].

The second challenge for the intracellular amastigote is to scavenge all its essential micro- and macronutrients to fuel growth and replication and adapt its cellular physiology to this new environment. *Leishmania* parasites are auxotrophic for purines, some amino acids, biotin, pterins, folic acid, pantothenate, pyridoxine, riboflavin, nicotinate and haem[13,14] and require other trace elements (including $Mg^{2+}$, $Ca^{2+}$, $Zn^{2+}$, $Mn^{2+}$ and $Fe^{2+}$[14]), all of which need to be salvaged. The PV contains a varying supply of nutrients derived from host cell digestive processes[13,15]. Some of the transporters involved in nutrient uptake have been identified, and some are extensively characterised, including multiple transporters for folate/biopterin[16], nucleosides[17], iron[18,19], haem[20], magnesium[21] and glucose[22]. Following a metabolic switch, termed stringent response, the slow-growing amastigotes utilise glucose more effectively, compared to promastigotes, without secreting partially oxidised glucose products acetate, alanine or succinate[23]. Amastigotes reduce uptake of glutamate and glutamine (relying on the Krebs cycle for de novo synthesis) and they scavenge fatty acids from the PV, which are actively catabolised in the parasite mitochondrion by β-oxidation[23]. The arginine transporter AAP3 is upregulated in amastigotes to counteract the host cell's attempt to deprive the pathogen of arginine[24]. Iron is another key micronutrient at the interface between pathogen and host cell. Surface and intracellular iron transporters have been implicated in *Leishmania* parasite differentiation and amastigote survival[18,19,21].

Organellar transporters also contribute to *Leishmania's* virulence, one example being the synthesis of lipophosphoglycan, which depends on nucleotide sugar transporters located in the Golgi[25]. Finally, membrane transporters are also conduits for the uptake and extrusion of anti-parasitic compounds, with well-characterised examples including the uptake of the drug miltefosine via the miltefosine transporter, a phospholipid-transporting P-type ATPase[26], and trivalent antimony via the aquaglyceroporin AQP1[27].

Proteins that mediate transport of substances across lipid bilayers are a structurally and functionally diverse cohort, with dissimilar architectures, ranging from single transmembrane monomers to large multimeric protein complexes. The Transporter Classification Database TCDB[28-30] and TransportDB 2.0[31] provide a classification system for these proteins based on structure and function. Comparing uncharacterised transporters against TCDB allows for classifications into families but in most cases no definitive prediction of transporter function is possible.

There are just over 30 transporters for which gene deletion attempts have been reported in the literature for any *Leishmania* sp. (see[32] and Supplementary Data 1). Many of these studies focused on transporters for sugars, nucleobases, folate, iron and heme and transporters implicated in drug resistance. For the majority of *Leishmania* membrane transporters, their contribution to promastigote and amastigote fitness remains untested. Here, we conduct a systematic loss-of-function screen across the whole *Leishmania* "transportome" of 312 putative transporter encoding genes, including predicted surface-localised proteins as well as organellar transporters. Using a CRISPR-Cas9 mediated gene deletion and barcoding strategy[33,34] we successfully generated 188 null mutants and another 81 mutant cell lines that still retained a copy of the target gene (see Supplementary Data 1). The mutants' fitness was tested in promastigotes by assessing their growth in vitro, and in amastigotes in human macrophages in vitro and in a mouse model of infection in vivo. This study identified numerous loss-of-fitness and some gain-of-fitness phenotypes in transporter deletion mutants in both parasite forms. It also identified a cohort of mutants that showed conditional essentiality only in amastigotes, with 17 transporter gene deletions exhibiting a significant loss of fitness both in cultured macrophages and mice and another 20 only in mice. These data show that the amastigotes are particularly vulnerable to the loss of proton pumps and several other transporters that merit further study.

## Results

### Identification and composition of the *L. mexicana* transportome
A combination of database queries and literature searches were performed to collate an inventory of the *L. mexicana* "transportome" comprising channels, carriers and primary active transporters for small molecules, ions and solutes. The *L. mexicana* genome annotation[35,36] was searched with relevant keywords and the resulting list was filtered for proteins that had relevant protein domains, sequence similarity to transporter proteins listed in TCDB or experimental evidence of transporter function. Components of transport systems involved in protein translocation, intraflagellar transport, nuclear pores and components of intraorganellar membrane contact sites were excluded from this study. This resulted in a set of 312 putative membrane transporter proteins, constituting approximately 3.8% of the *L. mexicana* proteome (Fig. 1, Supplementary Data 1), a value on par with the TransportDB 2.0 predictions for the membrane transporter repertoire in other single- and multi-cellular eukaryotic organisms (e.g. *Homo sapiens* 1.8%; *Caenorhabditis elegans* 2.5%; *Chlamydomonas reinhardtii* 2.8%; *Trypanosoma brucei* 3.9%; *Plasmodium falciparum* 2.3%)[31]. Blast searches of TCDB were conducted and the results were used to assign the *L. mexicana* proteins to superfamilies according to the Transport Classification (TC) system[29]. This identified hits in 49 different superfamilies with a range between one and 53 *L. mexicana* proteins identified as putative members of each family. Examining different modes of transport, we found 34 alpha-type channels (TCDB 1.A; 12 different families) and two beta-barrel porins (TCDB 1.B; mitochondrial porins, also known as voltage-dependent anion-selective channel [VDAC]), 184 proteins of secondary carriers (TCDB 2.A; 30 different families) and 91 proteins of primary active transporters (TCDB 3.A; including 53 ATP-binding Cassette [ABC] proteins, 19 V-type and A-type ATPase [F-ATPase] proteins, and 17 P-type ATPase [P-ATPase] proteins) and one auxiliary transport protein (TCDB 8.A) (Fig. 2A, Supplementary Data 1). Note that these numbers count individual proteins, some of which are subunits of multi-protein complexes and include some near-identical copies of gene arrays. For 73% of these gene products, we found

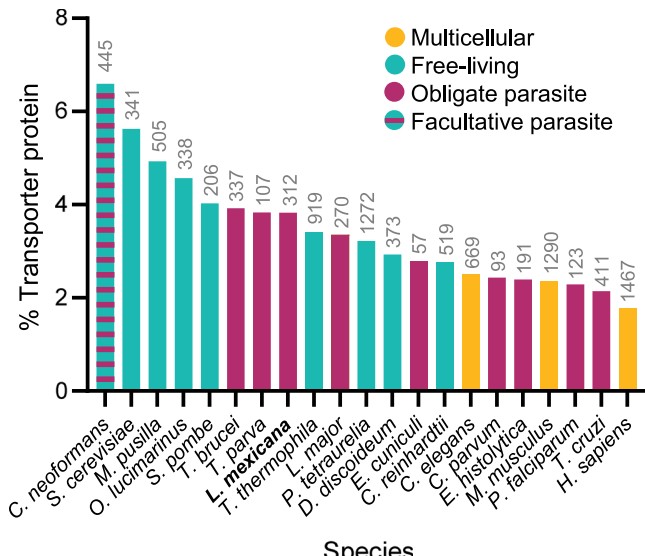

**Fig. 1 | Proportion of membrane transporter proteins in the proteomes of selected eukaryotes.** The bars indicate the percentage of membrane transporter proteins in the proteomes of selected parasitic and non-parasitic eukaryotes; the predicted number of transporters is written above each bar. TritrypDB[36] was used as reference for the number of predicted *L. mexicana* proteins and the number of *L. mexicana* transporters was determined in the present study. For all other species, the number of entries in the UniProtKB (Swiss-Prot and TrEMBL[88]) was used as a measure total protein number, and the number of transporters was taken from TransportDB 2.0[31]. Source Data are provided as a Source Data file.

published evidence of experimental characterisation in a trypanosomatid species or bioinformatics analysis, giving varying degrees of certainty about their specific functions (Supplementary Data 1). This leaves over one hundred *Leishmania* transporter proteins for which there has been little analysis beyond the assignment to TCDB families. Overall, the specific contribution to the fitness of *Leishmania* parasites has only been tested for a very small number of transporter proteins.

## Gene deletion yielded 188 null mutants
To study loss-of-function phenotypes, each of these 312 putative transporter genes was targeted for deletion in promastigotes, using a CRSIPR-Cas9-based method for gene replacement by drug-selectable donor DNA cassettes[33], yielding 301 viable mutant populations resistant to both selection drugs (Supplementary Data 1). Diagnostic PCR tests (Supplementary Fig. 1) confirmed the loss of the target gene for 188 of these (null mutants) while 81 still retained a copy of the target gene (referred to as 'incomplete knockouts') (Fig. 2B; Supplementary Data 2). The 11 genes, for which no viable mutants were recovered after at least two attempts, were then targeted using only a single drug resistance cassette (Puromycin). With this approach, 11 mutant cell populations were successfully recovered, providing technical validation for the targeting constructs. As expected, the recovered cell populations had integrated the puromycin cassette, but all had retained copies of the target gene ('incomplete knockouts'). For 32 cell lines, one or more of the controls in the diagnostic PCRs did not show the expected band and therefore the mutant genotype remained uncertain. For seven cell lines designated as null mutants, whole genome sequencing (WGS) was performed additionally to the diagnostic PCR, to verify the loss of the target gene (including two that had previously been reported to be refractory to deletion) and analyse chromosome ploidy. In all seven genomes, alignment of the Illumina reads from the mutant genome against the reference genome showed precise excision of the target gene in the mutants (Supplementary Fig. 2). Chromosome ploidy remained mostly constant in the mutants

compared to the parental genome, with a few exceptions, such as a copy number increase for chromosome 32 in the Δ*LmxM.31.2660* mutant (lacking a putative lysine transporter) and for chromosome 6 in Δ*MIT1 (LmxM.08_29.2780)* (Supplementary Fig. 3). None of the ploidy changes affected the chromosome on which the target gene was located. Whether these changes in chromosome ploidy were stochastic events or causally linked to the specific gene deletions cannot be inferred from these data.

Overall, the vast majority of the 188 confirmed deletion mutants generated here are novel and provide a population of viable mutant promastigote forms which could be subjected to phenotype analysis under different conditions.

## Deletion of genes arranged in tandem arrays
A closer examination of the genomic organisation of transporter genes identified 113 genes in 41 arrays of tandemly arranged transporter genes; these included 62 genes in dispersed arrays with similar genes on multiple chromosomes, 19 genes in mixed arrays of transporters belonging to different families and 32 genes in 16 unique arrays of similar genes confined to a single locus (Supplementary Fig. 4A; Supplementary Data 3). Since remaining genes could possibly compensate for the loss of individual members from an array, gene deletions strategies were designed to remove whole arrays (Supplementary Fig. 4B–D). We selected eight arrays where the deletions of individual genes from the array had all been successful, containing respectively Voltage-gated Ion Channel (VIC), Mitochondrial Carrier (MC), Cyclin M Mg²⁺ Exporter (CNNM), Major Facilitator Superfamily (MFS) and Amino Acid/Auxin Permease (AAAP) family genes. Complete array deletion was confirmed for one MC array (*LmxM.18.1290* and *LmxM.18.1300*; only 30% protein sequence identity) and one AAAP array (*LmxM.34.5350* and *LmxM.34.5360*; 52% protein sequence identity) (Supplementary Data 3, Supplementary Fig. 5).

## Transporter protein family encoding genes refractory to deletion
Redundancy could occur at the level of superfamily members beyond gene arrays. Across the 49 superfamilies of membrane transporter encoding genes (Fig. 2C), many genes from larger multi- and single copy families were dispensable for in vitro promastigote survival. Most of their substrates are not known, and therefore we cannot distinguish in this analysis between transporters whose function is not required in cultured promastigotes and transporters where there is a high level of redundancy between transporters from the same family. For the superfamilies with single members, viable null mutants were generated for nine: a mechanosensitive ion channel (MscS; Δ*LmxM.36.5770*); Golgi pH regulator (GPHR; Δ*LmxM.07.0330*); Ca²⁺:H⁺ antiporter-2 (CaCA2; Δ*LmxM.19.0310*); a K⁺ transporter (Trk; Δ*LmxM.34.0080*); a sulphate permease (SulP; Δ*LmxM.28.1690*); mitochondrial EF hand Ca²⁺ uniporter regulator (MICU; Δ*LmxM.07.0110*); Presenilin (Δ*LmxM.15.1530*), divalent anion:Na⁺ symporter (DASS; Δ*LmxM.28.2930*) and vacuolar iron transporter (VIT; Δ*LmxM.27.0210*), suggesting their functions are dispensable for promastigote growth under standard in vitro culture conditions. Conversely, there were six distinct families with only one or two proteins in *L. mexicana* for which no null mutants were obtained, namely the heme-responsive gene protein (HRG; LmxM.24.2230), Proton-dependent Oligopeptide Transporter (POT; LmxM.32.0710), Mitochondrial Inner Membrane K⁺/H⁺ and Ca²⁺/H⁺ Exchanger (Letm1; LmxM.08_29.0920), H⁺-translocating Pyrophosphatase (H⁺-PPase; LmxM.30.1220) and Glycoside-Pentoside-Hexuronide:Cation Symporter (GPH; LmxM.30.0040) or the two Acetate Uptake Transporters (AceTr; LmxM.03.0380 and LmxM.03.0400). These may serve indispensable functions in promastigote physiology.

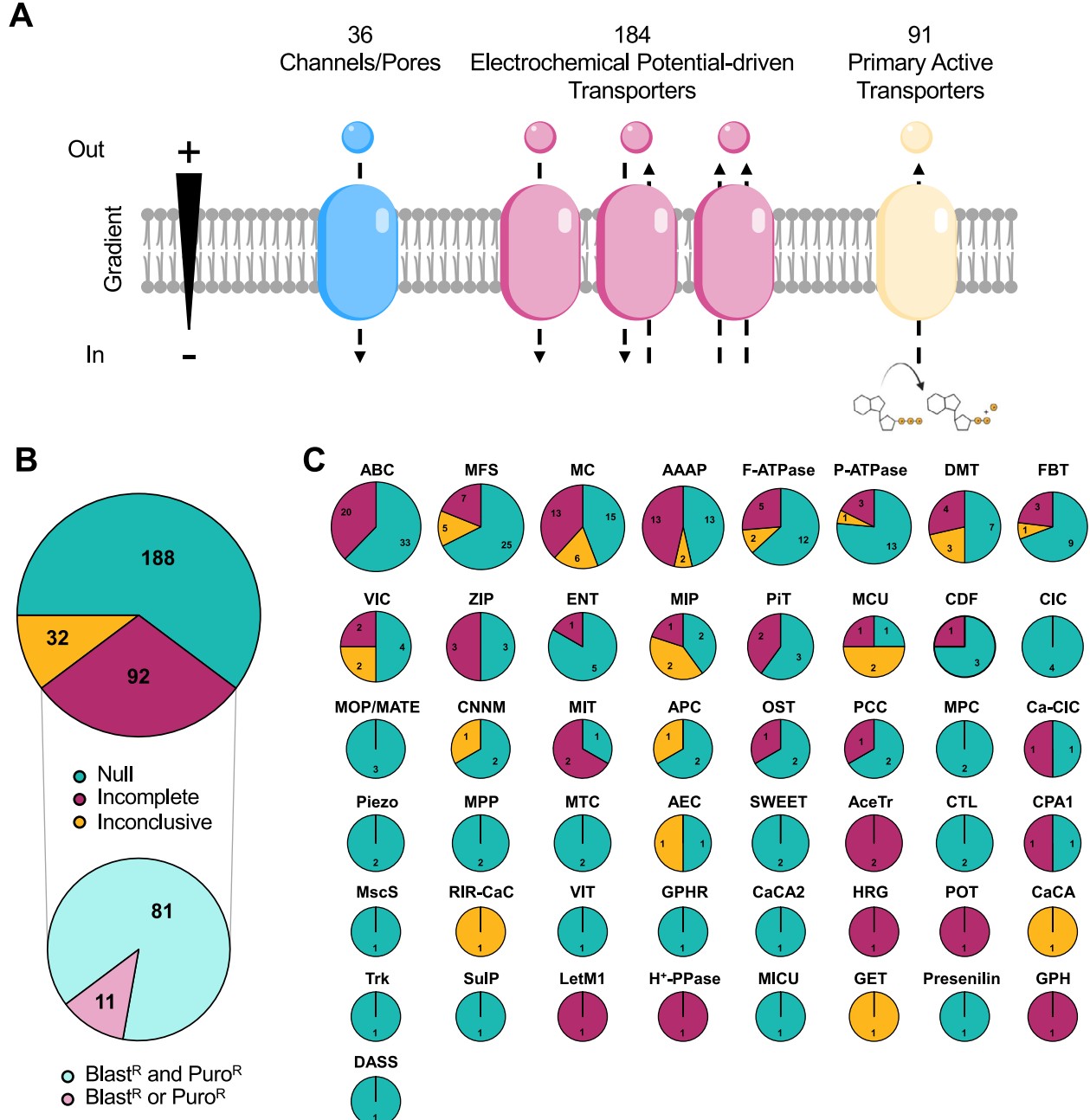

**Fig. 2 | Generation of the TransLeish knockout library. A** Overview of the number of predicted protein components of channels, carriers and pumps targeted for gene deletion in *L. mexicana*. **B** Summary of gene deletion results. The pie-charts show the numbers of successful deletions ('null'), incomplete deletions and inconclusive results. The category 'incomplete deletions' contained cell populations that were resistant to both selection drugs (Blasticidin and Puromycin) and populations that were only recovered from transfections with a single drug marker (Blasticidin or Puromycin). **C** Summary of gene deletion results separated into TCDB families. The KO status is indicated as for panel **B** and the numbers show how many proteins of each family are in each category. Larger gene families are depicted as larger circles. The source data are provided in Supplementary Data 1.

## Identification of transporter deletions that affect growth fitness of cultured promastigotes

The relative fitness of the viable transporter knockout mutants was tested next, by combining the mutants into pools and assessing their growth as promastigote forms in a standard culture medium. These pools contained 264 barcoded cell lines: five barcoded parental control lines (SBL1-5), 251 mutants from the TransLeish library, Δ*Ros3* (a subunit of the miltefosine transporter, *LmxM.31.0510*) and three non-transporter knockout mutants with known phenotypes [Δ*dihydroorotase* (*LmxM.16.0580*, slow growth), Δ*IFT88* (*LmxM.27.1130*), very slow

growth, no flagellum), Δ*MBO2* (*LmxM.33.2480*), normal growth, uncoordinated movement)], and four non-transporter knockout mutants with unknown phenotypes (Δ*LmxM.18.0610*, ΔLmxM.*30.2740*, Δ*LmxM.28.2410* and Δ*LmxM.15.0240*). These pools were grown in M199 culture medium, in three separate flasks, for two days. DNA samples were taken at the start of the experiment (0 h) and after 24 h (Fig. 3A) to quantify the abundance of each barcode at each timepoint (Supplementary Fig. 6A).

Fitness scores were calculated from barcode proportions at a given time point compared to the starting pool (Fig. 3B,

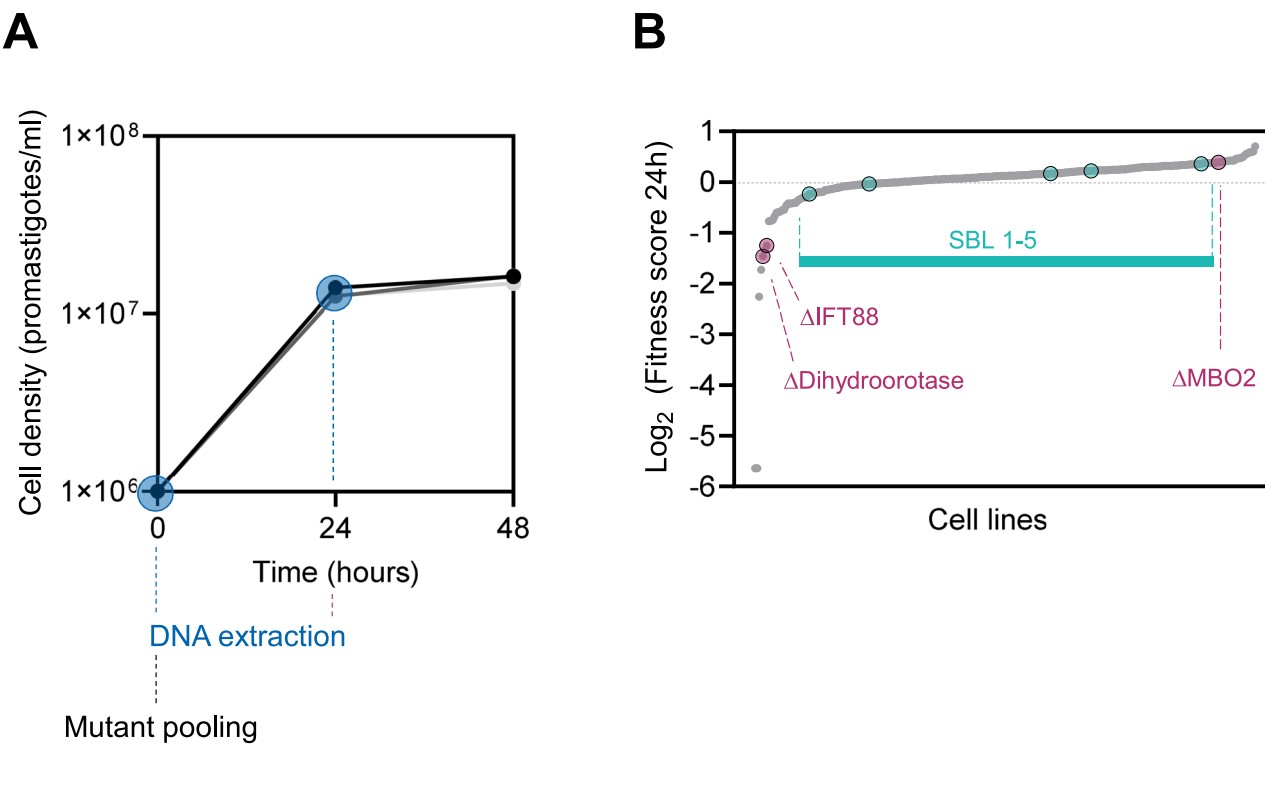

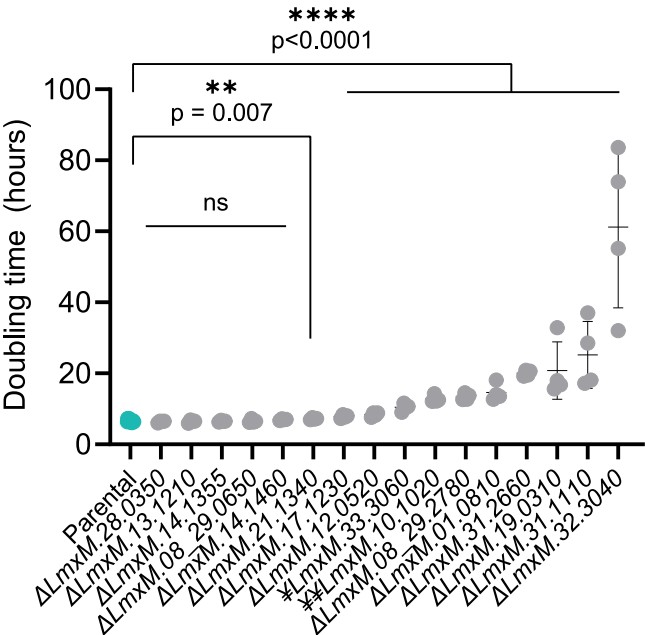

**Fig. 3 | Fitness of promastigote mutants in vitro. A** Growth curve of the mutant pool. Measurements for each of the three replicate samples are plotted individually. DNA was sampled at the time-points highlighted by a blue circle. Source Data are provided as a Source Data file. **B** Small grey dots indicate TransLeish mutant cell lines, ranked in order of their fitness score, from lowest to highest. Larger dots show parental control cell lines (SBL1-5) and control mutants, as labelled; The source data are provided in Supplementary Data 4. **C** Doubling times of the parental control line (turquoise dots), the slow growing control knockout cell line Δ*IFT88* (*LmxM.17.1230*) and selected transporter knockout mutants are plotted. Each dot represents the average doubling time of three replicates for a 24h-interval. The bars indicate mean and standard deviation for each cell line. Source Data are provided as

a Source Data file. Asterisks indicate mutants with a significantly increased doubling time compared to the parental (unpaired two-tailed t-test): Δ*LmxM.28.0350 (Sweet2)* p = 0.4019, Δ *LmxM.13.1210 (NT3)* p = 0.5612, Δ *LmxM.14.1355 (FT)* p = 0.7030, Δ *LmxM.08_29.0650 (Pho89)* p = 0.9338, Δ *LmxM.14.1460 (MPC2)* p = 0.0737, Δ LmxM.21.1340 (*V-ATPase V₁ H*) p = 0.0077, Δ *LmxM.17.1230 (P-ATPase)* p = 0.000087, Δ*LmxM.12.0520 (V-ATPAse V₁ F)* p = 6 x 10⁻⁰⁶, ¥*LmxM.33.3060 (MCP6)* p = 2 × 10⁻⁰⁸, ¥¥*LmxM.10.1020 (MCP19)* p = 2.4 × 10⁻¹¹, Δ*LmxM.08_29.2780 (MIT1)* p = 3.2×10⁻¹², Δ*LmxM.01.0810 (CAKC)* p = 9.9 × 10⁻⁹, Δ *LmxM.31.2660 (AAP7)* p < 10⁻¹⁵, Δ*LmxM.19.0310 (GDT1)* p = 1.3×10⁻⁵, Δ *LmxM.31.1110 (MCP2)* p = 3.4 × 10⁻⁶, Δ*LmxM.32.3040 (ABCI3)* p = 3.4 × 10⁻⁷. Mutants where gene deletion was inconclusive (¥) or incomplete (¥¥) are indicated.

Supplementary Data 4). None of the mutants showed enhanced fitness in the promastigote in vitro pool (score above 2 and $p < 0.05$). Seven mutants showed reduced fitness (score below 0.5 and $p < 0.05$), including the control lines $\Delta IFT88$ and $\Delta dihydroorotase$. Of the transporter null mutants, $\Delta MIT1$ (LmxM.08_29.2780), a mitochondrial pyruvate carrier-like protein deletion ($\Delta LmxM.31.1110$) and a putative lysine transporter deletion ($\Delta LmxM.31.2660$) showed significantly reduced growth fitness in vitro. To test the assumption that the fitness scores were linked to the growth rates of these mutants, their doubling times were measured individually, alongside selected mutants for which slow growth was noticed during the selection process. This confirmed that they grew significantly slower than the parental control lines (Fig. 3C). Three null mutants ($\Delta ABCI3$, LmxM.32.3040; $\Delta CAKC$, LmxM.01.0810 and $\Delta LmxM.34.5150$, a putative biopterin transporter) and three incomplete knockout mutants (LmxM.10.1020, LmxM.21.1690 and LmxM.30.0320) had zero read counts in some or all of the initial pool samples therefore their growth fitness could not be inferred with confidence from the pooled assay. Individual growth rate measurements that were conducted for three of these lines ($\Delta ABCI3$, $\Delta CAKC$ and the LmxM.10.1020 mutant) showed that their doubling times were significantly longer than for the controls (Fig. 3C), which may have led to their underrepresentation in the initial pool. Overall, only a small number of viable mutants showed significant growth defects as promastigotes and most grew at a rate similar to the parental line. This reflects the selection exerted on the mutants following gene deletion, where proliferation in standard laboratory culture medium over several weeks was a necessary condition for the mutants to progress to the pooled screens.

## Forty transporter deletion mutants showed a loss-of-fitness phenotype in vivo

In their life cycle, *Leishmania* parasites experience a profound change in milieu when they enter the mammalian phagocyte and differentiate to the amastigote form. The relative fitness of transporter knockout mutants in amastigote stages was tested in two models, in a mouse footpad infection model in vivo, and in human macrophages in vitro. For these experiments, the transporter mutants were included in a larger pool of 335 distinct cell lines, comprising of 254 transporter mutants, 61 knockouts of non-transporter genes (see methods), of which 5 served as control cell lines; $\Delta IFT88$ (LmxM.27.1130; very slow promastigote growth, avirulent), $\Delta Kharon1$ (LmxM.36.5850; normal promastigote growth, avirulent[37]), $\Delta BBS2$ (LmxM.29.0590; normal promastigote growth, avirulent), $\Delta PMM$ (LmxM.36.1960; normal promastigote growth, avirulent[38]), $\Delta GDP\text{-}MP$ (LmxM.23.0110; normal promastigote growth, avirulent[38]) and 20 barcoded parental control lines (SBL1-20) mixed into the pool at defined ratios from 1:1 (equal to each mutant) to 1:32 (32 times less). These cells were grown in M199 culture medium for 4 days and then stationary phase parasites were used to infect either human induced pluripotent stem cell derived macrophages (hiPSC-Mac) or BALB/c mice. DNA samples were taken from the pool before infection (0 h), from infected hiPSC-Mac at 3 h, 24 h, 48 h and 120 h post-infection and from infected mice at 72 h, 3 weeks (504 h) and 6 weeks (1008 h) post-infection (Fig. 4A) and a fitness score of the *Leishmania* mutants was calculated for each time point from their barcode proportions (Fig. 4B–H, Supplementary Data 4).

The fitness scores of the parasite lines tested in mice spanned a larger range compared to the lines tested in macrophage infections in vitro. The fitness scores for most of the parental cell lines remained near 1, indicating no change in barcode proportion over time, except for four parental lines seeded in the initial pool at a concentration of 1:16 or lower (Supplementary Fig. 7). This indicates where the detection limit for low abundance cell lines is likely to lie.

Overall, the majority of deletions assessed in this screen had little impact on the fitness of *L. mexicana* amastigotes in hiPSC-Mac over the observed 5-day time period, with fitness scores remaining in a range between 0.5 and 2. Looking for mutants that became significantly depleted under the tested conditions, we found the biggest differences at the final time-point (120 h), where 28 mutants presented with a fitness score below 0.5 ($p < 0.05$), including three of the control mutants ($\Delta GDP\text{-}MP$, $\Delta PMM$, and $\Delta Kharon$) and several proton pump mutants (subunits of the V-ATPase and a P-type H$^+$-ATPase). The lowest fitness scores were measured for $\Delta ABCI3$, $\Delta CACK$ and cells lacking a mitochondrial pyruvate carrier-like protein ($\Delta LmxM.31.1110$). On the other end of the range, six cell lines presented a fitness score above 2 ($p < 0.05$). Only two of these, $\Delta LmxM.32.1860$, lacking the ortholog of the glycosomal transporter GAT2[39] and $\Delta LmxM.01.0440$ (uncharacterised MFS transporter) also showed an enhanced fitness score in samples from 3-week infected mice, albeit without passing the test for statistical significance.

The largest effects were observed in the mouse model in vivo. As in the hiPSC-Mac infections, the majority of assessed gene deletions did not alter the fitness of the *L. mexicana* amastigotes in mice, compared to the parental line, but for a substantial minority (16) we measured a severe loss of fitness and for some a gain of fitness. The biggest spread of fitness-values was observed at the final time-point, 6 weeks post inoculation (Fig. 4H), where a total of 48 cell lines had a fitness score below 0.5 ($p < 0.05$) and one cell line presented a score above 2 ($p < 0.05$). All five of the control mutants ($\Delta GDP\text{-}MP$, $\Delta PMM$, $\Delta Kharon1$, $\Delta BBS2$ and $\Delta IFT88$) showed the expected in vivo fitness reduction below 0.5 ($p < 0.05$). A comparison of the data from the hiPSC-Mac and mouse infections identified 17 transporter gene deletion mutants that consistently returned low fitness scores in all samples taken after 120 h in macrophages in vitro, and 3 weeks and 6 weeks in vivo in the mouse footpad and another 11 in both the 3-week and 6-week mouse samples (Fig. 5A, B). The $\Delta GDP\text{-}MP$, $\Delta PMM$ and $\Delta Kharon1$ mutants in this group were known to have reduced virulence in amastigotes. The mutant fitness data suggest conditional essentiality for a number of diverse transporters in amastigotes across infection models: the ABC transporters LABCG5 (LmxM.23.0380; shown to be involved in heme salvage in *L. donovani*[40]), ABCI3 (LmxM.32.3040) and LmxM.33.0670, a calcium-activated potassium channel (CAKC; LmxM.01.0810), three mitochondrial carrier proteins (MCP2/MCP22 LmxM.31.1110, LmxM.08_29.2780, MCP6 LmxM.33.3060) and an organic solute transporter (OST) family protein (LmxM.36.6690) and one MFS protein of the PAD surface transporter family (LmxM.30.3170). Strikingly, 11 of the 20 transporter mutants that showed significant loss of fitness after 120 h in macrophages and ≥3 weeks in mice, carried deletions of proton pump proteins, namely ten subunits of the Vacuolar H$^+$ ATPase (V-ATPase) and one P-type H$^+$ ATPase.

## The *L. mexicana* V-ATPase pump is essential for survival inside the mammalian host cell and plays a role in pH tolerance

V-ATPases are highly conserved multi-subunit rotary pumps responsible for the acidification of numerous organelles and contributing to cellular pH homoeostasis. There are 17 homologues in the *L. mexicana* genome of the conserved yeast V-ATPase subunits, which were all targeted in the TransLeish knockout screen returning twelve confirmed null mutants, three 'incomplete' deletions and two inconclusive results (Fig. 6A, B). Under promastigote growth conditions, these mutants remained well represented within the pool (Fig. 6C). A comparison of the barcode trajectories for all V-ATPase mutants across all time-points in hiPSC-Mac and mice showed consistency: their proportions decreased over time, except $\Delta LmxM.31.0920$ (Fig. 7A, B). Interestingly, LmxM.31.0920 is a homologue of yeast Vph1p, which is one of two isoforms of the V$_o$ a subunit that is associated with distinct V-ATPase complexes with different sub-cellular localisations[41]. Deletion of the second V$_o$ a subunit isoform in *L. mexicana*, $\Delta LmxM.23.1510$, resulted in a significantly reduced fitness score in macrophages and mice.

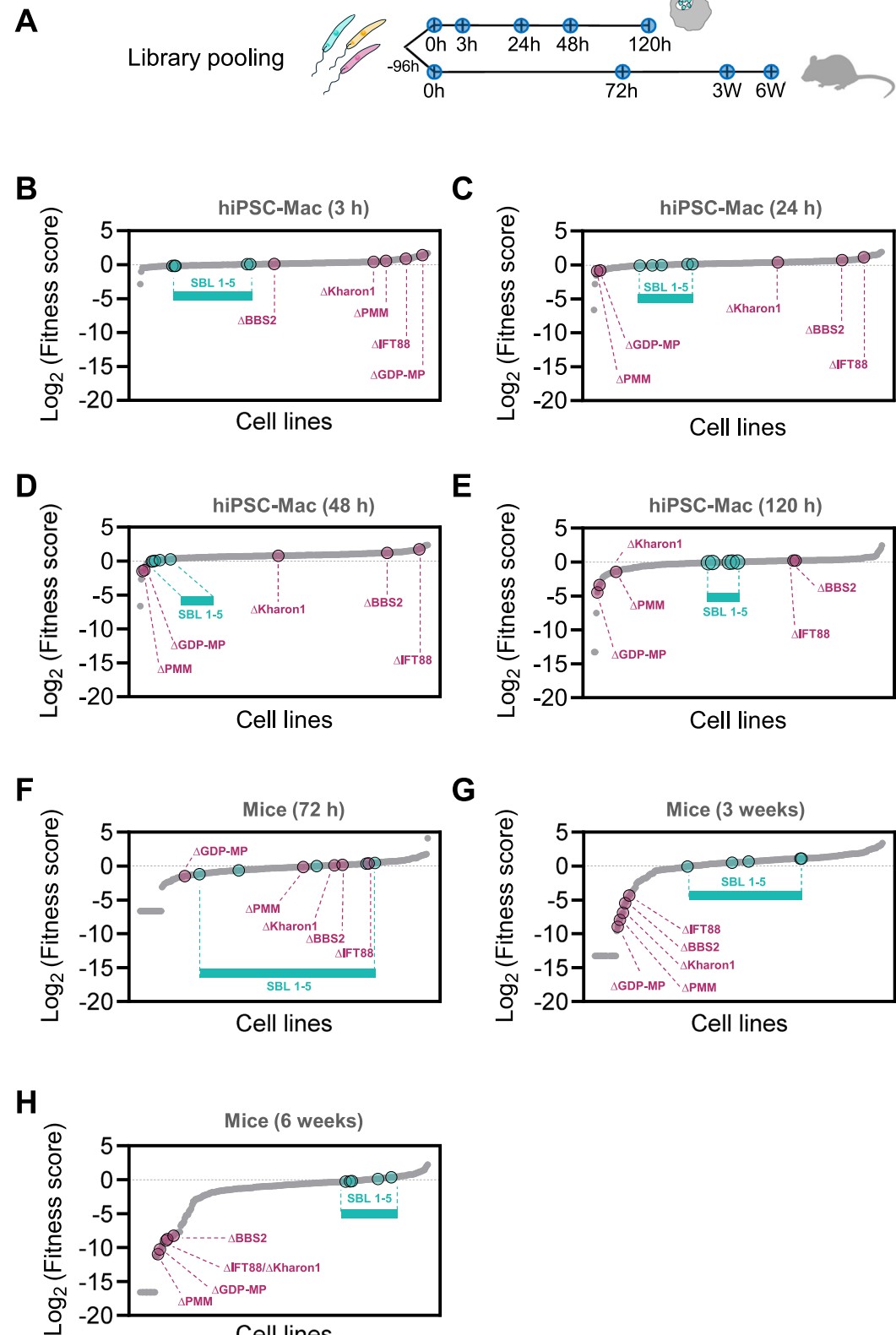

**Fig. 4 | Fitness of intracellular *Leishmania*, in macrophages and in mice.**
**A** Overview of the bar-seq timeline for knockout library pooling and infection of
hiPSC-Mac and mice. DNA sampling timepoints are marked by blue circles.
**B**–**E** Fitness scores of TransLeish mutant cell lines for each time point in hiPSC-Mac.

**F**–**H** Fitness scores of TransLeish mutant cell lines for each time point in mice. Small
grey dots represent the transporter mutants, ranked in order of their fitness score,
from lowest to highest. Larger dots show parental control cell lines (SBL1-5) and
control mutants, as labelled. The source data are provided in Supplementary Data 4.

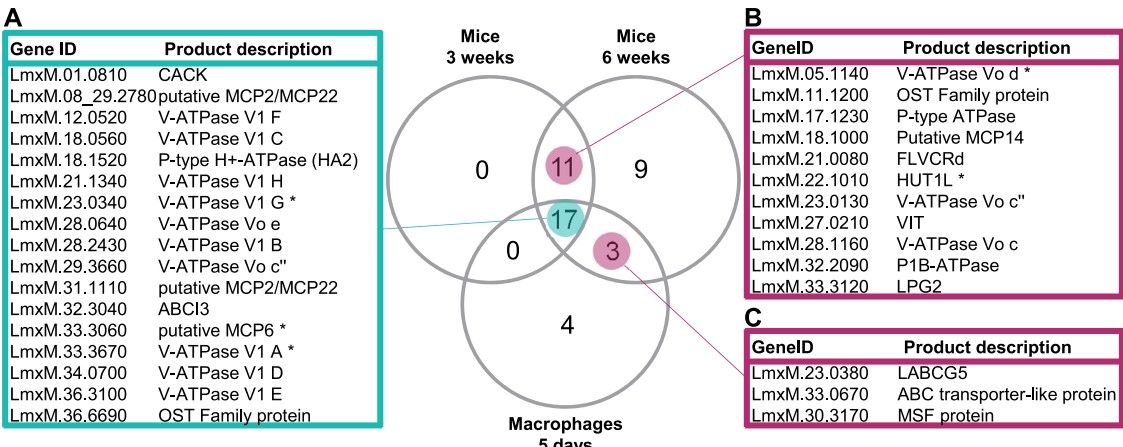

**Fig. 5 | Transporter mutants with significant loss of fitness in macrophages and mice.** Venn diagram of transporter mutants depleted in samples from macrophages infected for 5 days and mice infected for 3 and 6 weeks (fitness score below 0.5; p < 0.05). **A** Transporter mutants that showed a significant loss of fitness in all three conditions. **B** Transporter mutants that showed significant loss of fitness only in mice. **C** Mutants that showed significant loss of fitness at the final time points in macrophages and mice. Asterisks indicate mutants where gene deletion was incomplete. P-values were calculated using a Mann–Whitney U test; source data are provided in Supplementary Data 4.

A detrimental effect of V-ATPase gene deletions is compatible with their crucial role in regulating cellular pH in eukaryotic cells. What was surprising however was the discovery that loss of most V-ATPase subunits was compatible with promastigote viability and the deletions had only minor effects on promastigote growth in the pooled screen.

Since promastigotes were cultured at neutral pH while amastigotes reside in an acidic milieu of pH 5.5 or below[7], we hypothesised that the V-ATPase was conditionally essential in low external pH conditions. To assess pH tolerance of the parental cell line and the mutants, the growth rates of the parental cell line and two separate mutants, lacking V-ATPase subunits $V_1$ E or $V_1$ H, respectively, was compared in standard M199 medium at pH 7.4 and in M199 medium adjusted to pH 5.5. These experiments were conducted at 27°C, which prevents full differentiation to axenic amastigotes[42]. In one set of experiments, cells were passaged into fresh medium every 24 hours (Fig. 7C, D), in the second they were left to grow in the same flasks for 4 days continuously. At pH 7.4, doubling times ranged from 6.73 – 8.30 h, with little difference between the mutants and the control. At pH 5.5 the parental cell line maintained a doubling time ranging from 7.03 – 8.90 h. By contrast, the doubling time for both mutant cell lines progressively increased with each dilution in the acidic medium, from 7.03 to 32.63 h.

Together, these data suggest that the *L. mexicana* V-ATPase pump is required to maintain normal cellular physiology under low external pH conditions and consequently, amastigote survival inside the acidic parasitophorous vacuole is compromised in the deletion mutants.

V-ATPases are known to act as proton pumps for the acidification of cellular organelles and their localisation is regulated in response to different signals[43]. There are examples of V-ATPases acting on cell surfaces, including in the intracellular parasite *Toxoplasma gondii*, where it is thought to pump protons out of the cell under acid stress[44]. We examined the subcellular localisation of the *Leishmania* V-ATPase by tagging subunits $V_1$ G and $V_1$ H with mNeonGreen and imaging the fluorescence in live cells (Fig. 7E). For both subunits the same pattern was observed: The strongest signal was consistently seen in a crescent-shaped focus adjacent to the flagellar pocket, with weaker foci of variable intensities seen throughout the cell body. These localisation patterns were the same for promastigotes and axenic amastigotes (Fig. 7E). No signal was detected at the plasma membrane, suggesting the protective function of the V-ATPase under conditions of low external pH occurs at intracellular organelles and possibly at the flagellar pocket membrane.

## Discussion

Membrane transporter proteins are cellular gatekeepers for substance exchange between the cell and its environment and between membrane-bound subcellular structures and the cytoplasm. They are thus fundamental for maintaining cellular homoeostasis across contrasting microenvironments experienced by *Leishmania* parasites throughout their life cycle. In this study we tested for the first time the relative importance of 188 individual transporters in two distinct life cycle stages of this protozoan and discovered over forty transporter-encoding genes whose loss significantly reduced the fitness of intracellular amastigote forms in one or all of the tested conditions.

The vast majority of transporter null mutants generated in this study have never been reported before. Using the CRISPR-Cas9 method allowed for successful knockout of two genes that had been reported to be refractory to gene deletion by sequential homologous recombination in *L. amazonensis* and *L. major*, respectively, namely the mitochondrial iron transporter (MIT1, LmxM.08_29.2780[19]); and the mitochondrial ABC transporter ABCI3 (LmxM.32.3040[45]); Individual characterisation of these null mutants revealed a significant increase in promastigote doubling time, consistent with important functions for these transporters. It remains to be tested what compensatory mechanisms, if any, operate in these *L. mexicana* mutants to sustain vital functions. Our results further corroborate in *L. mexicana* previous findings from other *Leishmania* spp. that deletions of the *Leishmania* iron regulator 1 (LIR1; LmxM.21.1580) and the ferrous iron transporter (LIT1; LmxM.30.3060) are viable in promastigotes and LIR1 was confirmed to be important for virulence. Additionally, we discovered that loss of the putative vacuolar iron transporter LmxM.27.0210 (the only member of the VIT superfamily found in *Leishmania*) was tolerated in promastigotes but negatively impacted on the fitness of amastigotes in vivo (Fig. 5B). In agreement with previous studies, we were unable to generate gene deletions for a porphyrin transporter FLVCRb (LmxM.17.1430[46]) and heme transporter (LHR1; LmxM.24.2230[20]), underlining the crucial importance for heme salvage in this parasite that lacks all but the last three enzymes required for heme biosynthesis[47]. It was however possible to generate viable LABCG5 (ΔLmxM.23.0380) null mutants, thus removing a transporter reported to facilitate salvage of heme released from internalised haemoglobin[40]. Also refractory to deletion were the ABC transporter ABCB3 (LmxM.31.3080[48]) and magnesium transporter 2 (MIT/ LmxM.25.1090[21]) supporting previous reports that these may be essential genes.

**A**

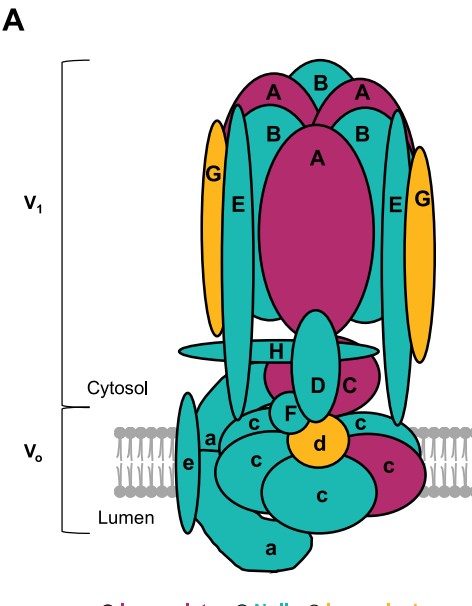

● **Incomplete** ● **Null** ● **Inconclusive**

**B**

| Gene ID | Sub-complex | Subunit | KO status |
|---|---|---|---|
| LmxM.33.3670 | $V_1$ | A | Incomplete |
| LmxM.28.2430 | $V_1$ | B | Complete |
| LmxM.18.0560 | $V_1$ | C | Incomplete |
| LmxM.34.0700 | $V_1$ | D | Complete |
| LmxM.36.3100 | $V_1$ | E | Complete |
| LmxM.12.0520 | $V_1$ | F | Complete |
| LmxM.23.0340 | $V_1$ | G | Inconclusive |
| LmxM.21.1340 | $V_1$ | H | Complete |
| LmxM.23.1510 | $V_0$ | a | Complete |
| LmxM.31.0920 | $V_0$ | a | Complete |
| LmxM.21.1800 | $V_0$ | c | Complete |
| LmxM.21.1790 | $V_0$ | c | Incomplete |
| LmxM.29.3660 | $V_0$ | c | Complete |
| LmxM.23.0130 | $V_0$ | c | Complete |
| LmxM.28.1160 | $V_0$ | c | Complete |
| LmxM.05.1140 | $V_0$ | d | Inconclusive |
| LmxM.28.0640 | $V_0$ | e | Complete |

**C**

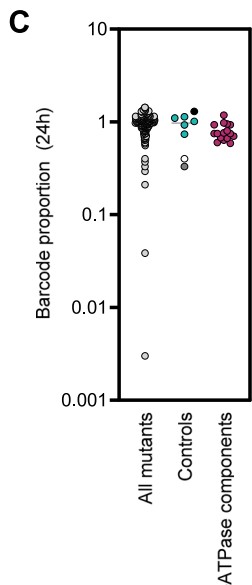

**Fig. 6 | *L. mexicana* V-ATPase mutants are viable as promastigotes. A** Schematic of V-ATPase subunits identified in *L. mexicana*, coloured by the genotype of the corresponding mutants. **B** Gene IDs of *L. mexicana* V-ATPase subunits. **C** Barcode proportions of mutants after 24 h of exponential growth in culture. Mutants were grouped into the following categories: 'All mutants' represent all non-V-ATPase transporter mutants in the pool (dark grey dots); 'Controls' represent the barcoded parental controls (SBL1-5, turquoise dots) and control mutants Δ*dihydroorotase* (light grey dot), Δ*IFT88* (white dot) and Δ*MBO2* (black dot); 'V-ATPase components' are shown as red dots. Each dot represents the average barcode proportion for one mutant cell line after 24 h (normalised to 0 h). The source data are provided in Supplementary Data 4.

There was a small number of genes for which knockouts had previously been reported that did not yield null mutants in this present study, encoding aquaglyceroporin 1 (*AQP1; LmxM.30.0020*), the iron/zinc permease (*LmxM.30.3070*) and the equilibrative nucleoside transporter (*BTN1; LmxM.22.0010*). Some of these failures may arise from technical limitations of this screen: For genes in tandem arrays, high sequence similarity led to inconclusive diagnostic PCRs, notably for the glucose transporter array (LmGT1-LmGT3) that was previously deleted in *L. mexicana*[22,49]. Similarly, for the tandem array of proline/alanine transporters AAP24, shown to be dispensable in *L. donovani*[50], deletion of one *L. mexicana* ortholog (*LmxM.10.0715*) was confirmed, but not for the second (*LmxM.10.0720*). The ABCG1 and ABCG2 array was demonstrated to be dispensable in *L. major*[51], yet gene attempts to delete the orthologs in *L. mexicana* (*LmxM.06.0080* and *LmxM.06.0090*) returned only 'incomplete' knockouts. Underestimation of gene copy number could also hamper deletion attempts. Although, using our double-drug selection strategy we were able to successfully delete 11 of the 35 transporter genes present on supernumerary chromosomes 3, 16, and 30; for the remainder a deletion could not be confirmed. Additional rounds of gene replacement with additional drug selection markers might have resulted in additional successful null mutant isolations. Taken together, these results identify at least 188 transporter proteins that are dispensable in cultured *L. mexicana* promastigotes, i.e. 60% of the transportome. Since highly similar gene sequences could have led to false positive diagnostic PCR results, and gene copy number variation poses technical challenges, the true number of dispensable proteins may even be higher, and it is important not to mistake a result of 'incomplete' knockout as definitive evidence for gene 'essentiality' without further experiments. Conversely, some of the transporters that were non-essential in cultured promastigotes may prove to be vital for passage through a sand fly.

Despite these caveats, the essential 'transportome' for promastigote in vitro must be a sub-set of the 40% of genes for which no null mutants were obtained. In the closely related kinetoplastid, *T. brucei*, growth fitness was assessed genome-wide for a library of RNAi knockdown cell lines grown as procyclics or bloodstream forms in vitro[52]. This included the 300 orthologs of the *L. mexicana* transporters identified in our study. For 102 (33%) of the *T. brucei* transporter orthologs, induction of RNAi resulted in cell line depletion in at least two of the tested conditions (≥1.5-fold change compared to the uninduced control; data viewed on TritrypDB.org[36]). Further systematic comparisons of transporter repertoires and mutant fitness across different kinetoplastid species and life cycle stages could support the identification of transporters that are universally required for kinetoplastid cell functions, and those that may be vital for the disease-causing life cycle stages. A systematic analysis of the transporter repertoire and essentiality in *T. cruzi*, which inhabits a different intracellular niche compared to *Leishmania* would be particularly interesting. Amongst other protozoan parasites, a combined analysis of all available reverse genetic experiments on *Plasmodium falciparum* and *P. berghei* concluded that 78% of *Plasmodium* sp. transporters were essential at some point in its life cycle[53]. Similar focused analyses of existing gene deletion screens of other intracellular parasites, e.g. *Toxoplasma*[54] and expanding phenotype analyses to more in vivo situations and varying environmental conditions will be key to unpicking the conditionally essential transporter repertoire for all of these parasite species in each of their forms.

The phenotyping of amastigote forms showed a dramatic depletion of proton pump mutants in samples taken from macrophages and mice. Only one of the P-type $H^+$ATPase genes, HA2, could be deleted in promastigotes, where no growth defect was apparent, but the mutants showed a severe fitness reduction in amastigotes. In *L. donovani*, HA1 was shown to be constitutively expressed while HA2 transcripts were expressed predominantly in amastigotes[55] supporting the notion of stage-specific functions for these genes. In both cases it is likely they function as typical single-subunit P-type H + -ATPases, which are found on the surface of plants, fungi and protists, but not mammalian cells.

The V-ATPases are highly conserved multi-subunit complexes whose proton pumping activity serves a variety of cellular functions, mostly leading to acidification of cell organelles, but occasionally also

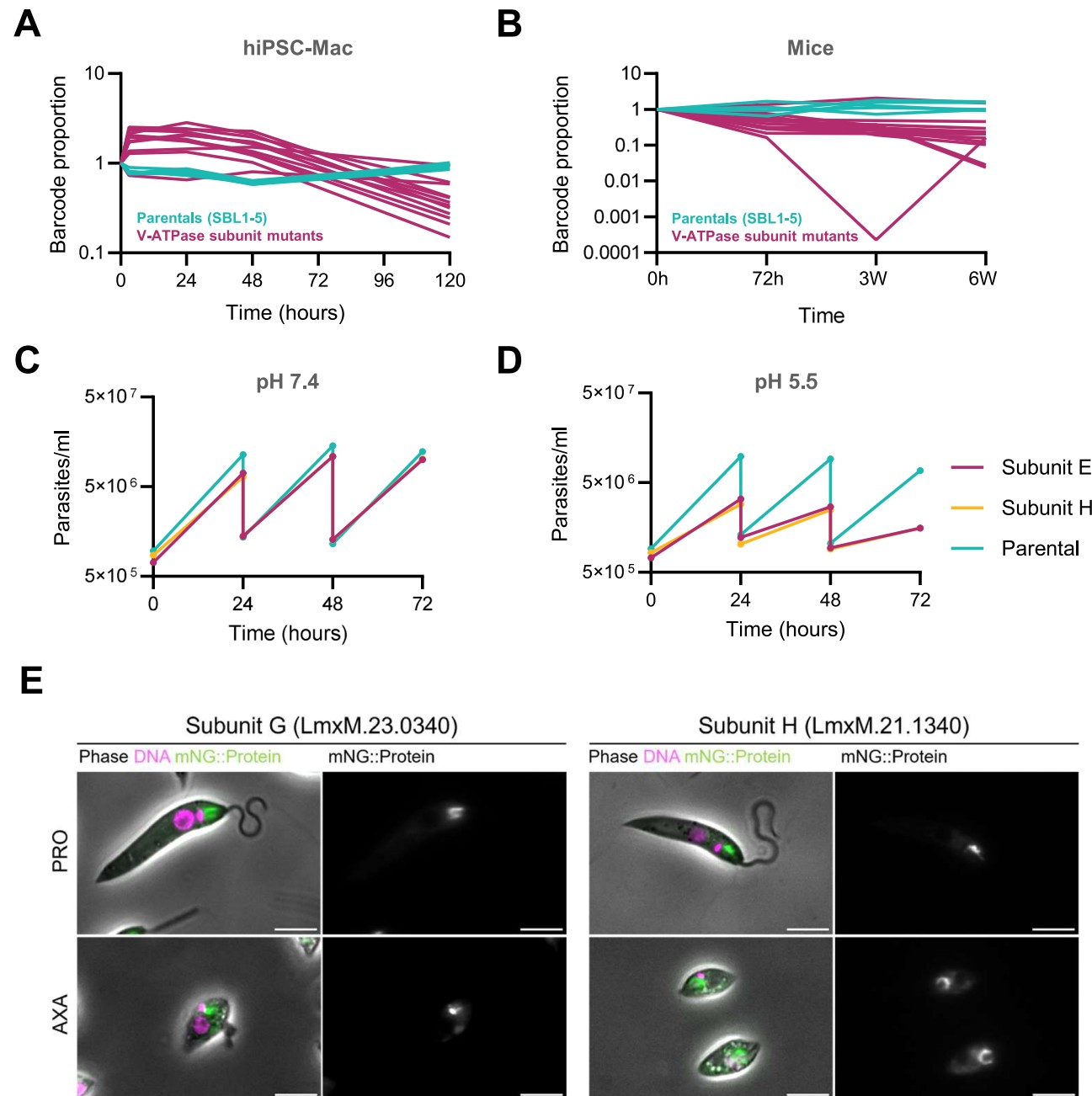

**Fig. 7 | V-ATPase mutants show significant loss of fitness in macrophages and mice and reduced growth rates under low external pH conditions. A** Barcode trajectories of barcoded parental controls and V-ATPase knockout mutants in macrophages. **B** Barcode trajectories in mice. **C** Growth of promastigote form V-ATPase deletion mutants $V_1$ E and $V_1$ H and the parental cell line in standard M199 medium at pH 7.4. Cells were diluted into fresh medium to a density of $1 \times 10^6$ cells/ml every 24 h. Each point represents the average cell density measured from three separate cultures. **D** Growth profiles as in C, in M199 medium adjusted to pH 5.5. **E** Fluorescence micrographs showing the localisation of mNeonGreen-tagged V- ATPase subunits $V_1$ G and $V_1$ H in promastigotes (PRO) and axenic amastigotes (AXA). For each subunit, the left panel shows the merged images from phase contrast illumination, and fluorescent signals from Hoechst-stained DNA (magenta) and the mNeonGreen fusion protein (green). The right panel shows the signal from the fusion protein only. The scale bar is 10 μm. The micrographs are representative of the mNG localisation results from two independent experiments tagging these genes. Source Data for **A** and **B** are in Supplementary Data 4. Source Data for **C**–**E** are provided as a Source Data file.

acting at the cell surface membrane of e.g. mammalian osteoclasts or tumour cells, reviewed in ref. 43. Protection against acid stress is a well-documented function for the V-ATPase of yeast, which emerged in screens of mutant libraries as amongst the key determinants in tolerance of strong and weak acids[56,57]. In the apicomplexan parasite *Toxoplasma*, the V-ATPase serves a dual role. In extracellular parasites, the V-ATPase at the parasite's plant-like vacuole protects the cells against ionic and osmotic stress, while intracellular parasites were found to position it at the plasma membrane, suggesting it may serve to pump protons out of the parasite cell to protect from acid stress[44]. Individual characterisation of the two *L. mexicana* mutants lacking subunits E and H (Δ*LmxM.30.3100* and Δ*LmxM.21.1340*, respectively), showed that in neutral pH, no significant differences in growth were observed when compared to the parental control but tolerance to acidic pH was much reduced. These results point to a protective role of the *Leishmania* V-ATPase in acidic environments. One simple model could be for the V-ATPase to extrude protons directly at the cell surface. Our data argues against a re-location of the V-ATPase to the cell surface, but one

possibility that remains compatible with our localisation data is that cytosolic protons are extruded via the flagellar pocket. Or data also supports a function of the *L. mexicana* V-ATPase in organellar homeostasis: Both in promastigotes and amastigotes, the tagged *Leishmania* V-ATPase subunits (Fig. 7) consistently localised to internal foci, suggestive of endocytic organelles, and compatible with a fraction also localising to acidocalcisomes. This is consistent with other reports on the localisation of V-ATPase subunits in trypanosomatids, using proteomics methods[58,59] or tagging and microscopy[60–62].

While acid sensitivity alone could explain the demise of amastigote V-ATPase mutants, there are indications that the role of the V-ATPase is more complex. Tolerance of stress conditions other than acid stress may require a functioning V-ATPase. Further detailed studies will be required to dissect the phenotype of the V-ATPase. One of the cellular processes requiring V-ATPase is autophagy[63], which is involved in the differentiation from *Leishmania* promastigotes to amastigotes[64]. Abolishing V-ATPase function may therefore already block parasite development before the cells are immersed in the acidic luminal contents of the PV, which could provide an alternative explanation for the depletion of these mutants in vivo. Endocytosis is yet another process for which kinetoplastid V-ATPases are important, as studied in *T. brucei*[65]. Assuming *Leishmania* V-ATPase has a similar function, impeding endocytosis, could also render the amastigotes less virulent. The fact that promastigote *L. mexicana* V-ATPase knockout mutants are viable allows for follow-up experimental dissection of these phenotypes.

The exquisite sensitivity of amastigotes towards the loss of proton pump function also marks them as candidates for potential drug targets to combat PV-dwelling pathogens. Encouraging results were obtained with Bafilomycin B1, a macrolide inhibitor of V-ATPase, which had an $IC_{50}$ of <1 nm against *L. donovani* and *T. cruzi* amastigotes[66], indicating amastigote themselves rely on a working V-ATPase, consistent with our genetic evidence. Modulation of the PV environment via host-cell dependent processes could offer additional routes to parasite killing. De Muylder and co-workers[67] showed that administration of the μ-opioid receptor antagonist naloxonazine to *Leishmania* infected macrophages inhibited intracellular parasite growth. This was linked to an upregulation of the host V-ATPase and increased volume of acidic vacuoles in the host cell. Combined administration of naloxonazine and the V-ATPase inhibitor concanamycin A restored normal infection levels.

We hypothesise that the P-type $H^+$ATPase and V-ATPase are both required for tolerance of an acidic extracellular milieu. Our working model is that the P-type $H^+$ATPase is primarily responsible for maintaining cellular pH homoeostasis in response to acidified environments by acting as a proton pump at the cell surface. The V-ATPase likely has more varied functions, and possibly multiple cellular locations. Besides the well-characterised activities of bafilomycins and concanamycin A, which inhibit all known eukaryotic V-ATPases, there are other inhibitors specific to certain species or isoforms[68]. Further optimisation may allow for targeted perturbation of host and/or parasite functions to eliminate the parasites from their intracellular niche.

## Methods

A list of reagents and resources used in this study is available in Supplementary Table 1.

### Experimental animals

All experiments were conducted according to the Animals (Scientific Procedures) Act of 1986, United Kingdom, and had approval from the University of York, Animal Welfare and Ethical Review Body (AWERB) committee (protocol number PP1651724). Eight-week-old female BALB/c mice were purchased from Charles River Laboratories and maintained in the pathogen-free facility at the University of York. All mice used in experiments were socially housed under a 12 h light/dark cycle and were only used in this experiment.

### Pluripotent stem cells

Three human induced pluripotent stem cell (hiPSC) lines (SFC840-03-03, SFC841-03-01, and SFC856-03-04) used in this study were previously derived from in-house grown dermal fibroblasts from three healthy human donors reprogrammed with Cytotune Sendai viruses (ThermoFisher). They have been published previously[69–71] and are deposited in EBiSC (STBCi026-A, STBCi044-A, STBCi063-A). They were maintained in the Stem Cell Facility at the James & Lillian Martin Centre for Stem Cell Research (Sir William Dunn School of Pathology, University of Oxford (United Kingdom), as QCed frozen banked stocks and cultured in mTeSR (StemCell Technologies) on Geltrex (Invitrogen) with minimal subsequent passaging (using 0.5 mM EDTA (Invitrogen) to ensure karyotypic integrity, as previously described[72]. Ethical approval for the derivation and use of these cells was obtained (Ethics Committee: National Health Service, Health Research Authority, NRES Committee South Central, Berkshire, UK, REC 10/H0505/71) with written informed consent by donors that specifically stated that their skin biopsies would be used for the derivation of pluripotent stem cell lines, and sharing of the cells with other researchers beyond the original deriving research team. This includes differentiation to cell types including immune cells, as in this study.

### *Leishmania* parasites

Promastigote forms of the *L. mexicana* cell line *L. mex* Cas9 T7[33] were grown in T25 cm$^2$ flasks at 28 °C or flat bottom well plates at 28 °C + 5% $CO_2$ in filter-sterilised M199 medium (Life Technologies) supplemented with 2.2 g/L $NaHCO_3$, 0.005% hemin, 40 mM 4-(2-Hydroxyethyl)piperazine-1-ethanesulfonic acid (HEPES) pH 7.4 and 10 % FCS. 50 μg/ml Nourseothricin Sulphate and 32 μg/ml Hygromycin B were added to the medium for the maintenance of the *sp*Cas9 and T7 RNA polymerase transgenes[33]. For the generation of axenic amastigotes, promastigotes were placed in M199 adjusted to pH 5.5 and incubated for 48 hours at 34°C with 5% $CO_2$.

### Transportome protein identification

Membrane transporter proteins included in the TransLeish library (Supplementary Data 1) were identified through iterative search strategies using sequence annotations, protein sequence domains and sequence homology with proteins in the transporter classification database TCDB, in combination with literature searches and manual curation of the final candidate list. Initially, the genome of *L. mexicana* MHOM/GT/2001/U1103 (TritrypDB[36]) was searched with the keywords *"transporter, transport, exchanger, permease, carrier, channel, porin, pump, symporter, antiporter, uniporter, porter, facilitator, efflux, ABC, ATP-binding cassette, P-glycoprotein, ATPase"*. Hits were filtered to include only proteins with domains indicative of transporter function, using Pfam domains and other sequence signatures integrated into the InterPro database[73]. In parallel, data on *Leishmania* transporters extracted from TransportDB 2.0[31] was merged with the initial candidate list. In a second round, the *L. mexicana* proteome was searched again through TritrypDB for additional proteins with relevant sequence signatures and new hits were added to the candidate list. Protein sequences from this list were used as queries for BLAST searches of TCDB using the default parameters. This allowed for the assignment of protein family descriptions and TCDB IDs. Protein sequences were scanned for the presence of trans-membrane domains by TMHMM (integrated in TritrypDB annotations) and CCTOP[74]. The refined and annotated list was then further validated and manually curated by comparing it against the published literature, which was also searched for information on proven or putative transporter

function (considering the *L. mexicana* proteins and any trypanosomatid orthologs identified through TritrypDB). Finally, components of transport systems involved in protein translocation, intraflagellar transport, nuclear pores, proteins of the electron transport chain and components of intraorganellar membrane contact sites were excluded from the list of candidate proteins, resulting in a predicted "transportome" of 312 proteins (Supplementary Data 1).

## Parasite growth curves

To determine the doubling time of slow growing mutants, supplemented M199 medium (M199) was seeded with $1 \times 10^5$ cells/ml, in three replicate cultures for each cell line. The cells were left to grow for 24 h at 28 °C. Cell density was measured after 24 h and cultures were diluted back to $1 \times 10^5$ cells/ml in fresh medium and left to grow again for 24 h. Cell growth was thus assessed four times on consecutive days, generating twelve cell counts in total for each cell line. For assessment of cell growth under different pH conditions, the pH of supplemented M199 medium was adjusted to either 7.4 or 5.5, using HCl 10 N solution. Cells were seeded at a starting density of $1 \times 10^6$ cells/ml and left to grow for up to 72 h at 27 °C with fresh dilutions every 24 h as above; each condition was assessed in three replicate cultures. Cell counts and cell volumes were determined with the CASY® cell counter (Cambridge Bioscience) using a 60 μm capillary and measurement range set between 2 and 10 μm.

## Macrophage cell culture

Human induced pluripotent stem cell (hiPSC) derived macrophages (hiPSC-Mac) were generated as previously described[72,75]. Briefly, iPSC from three different healthy human donors were expanded for one week, lifted with tryplE (Invitrogen), formed into uniform 10,000-cell embryoid bodies using Aggrewells (StemCell Technologies) in mTeSR with 1 mM Rock-inhibitor (Y27632; Abcam) and differentiated towards hemogenic endothelium with growth factors 50 ng/mL BMP4 (Invitrogen), 50 ng/mL VEGF (Invitrogen), and 20 ng/mL SCF (Miltenyi) with daily medium change for one week. The embryoid bodies were then transferred to T175 flasks in XVIVO-15 (Lonza) with 100 ng/mL M-CSF (Invitrogen) and 25 ng/mL IL-3 (Invitrogen) to induce myelopoiesis. Macrophage precursors were harvested from the supernatant, passed through a 40 μm cell strainer to exclude cell aggregates or embryoid bodies, centrifuged at 400 G, resuspended in macrophage medium and cultured in T25 flasks (Advanced DMEM/F-12 (Gibco), GlutaMAX 2 mM (Gibco), HEPES 15 mM (Gibco), human recombinant Insulin solution 5 μg/mL (Sigma), M-CSF 100 ng/mL (Invitrogen) Penicillin-Streptomycin 1% (Gibco)) for 7 days at 37 °C and 5% $CO_2$, with medium changes at day 3–4, to induce differentiation to ready-to-use adherent macrophages.

Twenty-four hour prior to infections, a total of $10^6$ hiPSC-Mac were seeded in the wells of flat-bottomed 6-well plates, medium was switched to fresh pre-warmed DMEM + 10 % FBS + 1 % pen-strep and cells were further incubated at 34 °C, 5% $CO_2$. Pools of barcoded *Leishmania* cell lines were generated as described in the section "Pooling of cells for bar-seq experiments". Stationary phase cultures of promastigote forms were used to infect the 34 °C-adapted hiPSC-Mac with a multiplicity of infection of 20 parasites to 1 macrophage. After 3 h cells were washed three times with medium to remove free parasites. Fresh medium was added to the washed infected hiPSC-Mac and cells were left to incubate at 34 °C, 5% $CO_2$ until DNA extraction.

## Animal work

To assess virulence and fitness of the null mutants in the pooled knockout library, three groups of six female BALB/c mice were infected subcutaneously at the left footpad with $2 \times 10^6$ promastigotes per animal. Presence or absence of lesion was observed, and footpads were measured weekly using a calliper. Animals were culled after 72 h, 3- and 6-weeks post inoculum for collection of the footpad.

## CRISPR-Cas9 gene knockouts and tagging

Gene deletions and tagging were done using the CRISPR-Cas9 method described in ref. 33. In brief, the *L. mex* Cas9 T7 cell line was transfected with a mix of four PCR amplicons: Two serving as transcription templates for sgRNAs, targeting sites upstream (5′) and downstream (3′) of the target gene ORF, the other two amplicons serving as donor DNA for DNA break repair, containing a blasticidin and puromycin resistance gene, respectively, flanked by 30 nt of sequence identical to the target locus and a unique 17-nt barcode together with the barcode flanking sequences GTGTATCGGATGTCAGTTGC and GTATAATGCAGACCTG CTGC[34]. The primer sequences for generating the PCR amplicons were taken from[76,77] and pTBlast and pTPuro served as template DNA for the amplification of donor DNA. For array deletions, an identical strategy was used, by replacing the gene array, with donor DNA cassettes that were amplified using the upstream forward primer for the first gene in the array and the downstream reverse primer for the last gene in the array. Parasite transfections were done on 96-well plates as described in ref. 78. Puromycin and blasticidin were added between 6 and 12 h after transfection and drug-resistant populations selected as described in ref. 78, with at least four passages before extraction of genomic DNA. Fitness screens were done using populations for which gene deletions were confirmed by diagnostic PCR, without further subcloning. For the expression of V-ATPase subunits with a fluorescent tag at the N-terminus, the gene encoding mNeonGreen (mNG) was inserted at the endogenous loci for *LmxM.21.1340* (V-ATPase Subunit H) and *LmxM.23.0340* (V-ATPase Subunit G) to create in-frame gene fusions, using pPLOT-mNG-Blasticidin for the generation of donor PCR products as described previously[33].

## Diagnostic PCR for knockout validation

Genomic DNA was extracted using the protocol from[79]. A diagnostic PCR was done to test for the loss of the target gene sequence in putative knockout cell lines, with *L. mex* Cas9 T7 gDNA serving as a control. A second PCR was performed on the mutant gDNA, amplifying a fragment of the blasticidin resistance gene with primers 518 F and 518 R. Finally, to test for presence of gDNA in samples extracted from mutants independently from the introduction of the drug resistance cassette, primers targeting genes *LmxM.34.5260* (*FAP174*), *LmxM.18.0610* or *LmxM.18.1620*, respectively, were used for PCR amplification. Primer sequences were taken from[80] and are listed in Supplementary Data 5.

## Pooling of cells for bar-seq experiments

The barcoded mutant and parental cell lines were combined such that the mixed pool contained similar numbers of each individual cell line: For the in vitro screen, a total of 252 transporter mutants, five barcoded parental lines (SBL1-5; barcodes introduced into the SSU locus[80]), and seven non-transporter knockout mutants, of which three acted as controls; Δ*IFT88* (*LmxM.27.1130*), Δ*dihydroorotase* (*LmxM.16.0580*) and Δ*MBO2* (*LmxM.33.2480*), were combined into a pool of $1 \times 10^6$ cells/ml. This pool was split into three equal aliquots, which were left to grow in separate flasks for 48 hours at 28 °C + 5% $CO_2$ in M199 medium. For iPSC-derived macrophage and mouse infections, a total of 254 transporter mutants (184 confirmed null mutants and 70 incomplete knockouts, see Supplementary Data 1), 57 knockout mutants of flagellar proteins, five barcoded parental lines (SBL1-5) and five control knockout mutants Δ*Kharon1* (*LmxM.36.5850*), Δ*BBS2* (*LmxM.29.0590*), Δ*IFT88* (LmxM.27.1130), Δ*PMM* (*LmxM.36.1960*) and Δ*GDP-MP* (*LmxM.23.0110*) were pooled in similar proportions. Fifteen additional barcoded parental lines (SBL6-20) were added to the pool at dilutions of 1:2 (SBL 6-8), 1:4 (SBL 9-11), 1:8 (SBL 12-14), 1:16 (SBL 15-17) and 1:32 (SBL 18-20), relative to the other mutants. To account for the slower growth of some mutants, which was observed during selection of the mutant lines, the *Leishmania* were first combined in three sub-pools, seeded at different initial cell

densities in M199 medium based on a qualitative assessment of their growth: 325 cell lines with "normal" growth, $1 \times 10^6$ cells/ml; six "slow" growing cell lines, $3 \times 10^6$ cells/ml; four "very slow" cell lines, $6 \times 10^6$ cells/ml. The three sub-pools were then left to grow for four days to stationary phase before combining the cells into a masterpool, from which replicates were prepared for DNA isolation (T0) and infection of macrophages or mice in 6 replicates.

## Sampling of DNA for sequencing

*Leishmania* promastigote genomic DNA was extracted from approximately $1 \times 10^7$ cells using the Qiagen DNeasy Blood & Tissue Kit according to the manufacturer's instructions, eluting in 40 µl of bi-distilled water (Ambion). For extraction of genomic DNA from infected macrophages, cells were scraped from the bottom of the wells, pelleted by centrifugation and total DNA was extracted, using the Qiagen DNeasy Blood & Tissue Kit as described above. DNA extraction from the mouse footpad lesions was performed using the Qiagen DNeasy Blood & Tissue Kit according to the manufacturer's instructions, except for an extended incubation in proteinase K solution and a final elution in 75 µl of TE buffer.

## Bar-seq library preparation and sequencing

For each DNA sample, the barcode region was amplified with custom designed p5 and p7 primers containing indexes for multiplexing and adapters for Illumina sequencing (The primers were synthesised at 0.2 µmol scale and PAGE purified by Life Technologies or by Microsynth.ch)[80].

For promastigote DNA samples, the quantity, purity and length of the total genomic DNA was assessed using a Thermo Fisher Scientific Qubit 4.0 fluorometer with the Qubit dsDNA HS Assay Kit (Thermo Fisher Scientific, Q32854), a DeNovix DS-11 FX spectrophotometer and an Agilent FEMTO Pulse System with a Genomic DNA 165 kb Kit (Agilent, FP-1002-0275), respectively. Bar-seq amplicon libraries were made via PCR as follows: 100 ng *Leishmania* gDNA, 1 µl of 10 mM dNTP Mix (R1091, Thermo Fisher Scientific), 1.25 µl of 4 µM P5 primer, 1.25 µl of 4 µM P7 primer, 1 µl of Platinum SuperFi II DNA Polymerase (12361010, Thermo Fisher Scientific), 10 µl of the 5X SuperFi II Buffer and nuclease free water in a total volume of 50 µL. A negative template control (NTC) was included at each PCR set-up. The following three-step protocol, was employed: an initial denaturation at 94 °C for 5 min, then 31 cycles of denaturation at 94 °C for 30 s, annealing at 60 °C for 30 s and extension at 72 °C 15 min. A final extension at 72 °C for 7 min was employed before a 4 °C hold. The CleanNGS kit (CNGS-0050, Clean NA) based on paramagnetic particle technology was used according to the user manual to purify each amplicon library. Thereafter, each library was evaluated using a Thermo Fisher Scientific Qubit 4.0 fluorometer with the Qubit dsDNA HS Assay Kit (Thermo Fisher Scientific, Q32854) and an Agilent Fragment Analyzer (Agilent) with a HS NGS Fragment Kit (Agilent, DNF-474), respectively. The bar-seq library shows a clear peak at 218 bp. Libraries were equimolar pooled into one pool. These library pools were further qPCR-based quantified using the JetSeq Library Quantification Lo-ROX kit (BIO-68029, Bioline) according to the manufacturer's guidelines. The amplicon library pools were diluted to 4 nM and spiked with 30 % PhiX Control v3 (illumina, FC-110-3001) and loaded at an on-plate concentration of 8 pM to reach a cluster density of ~1'000 K/mm2. The libraries were-paired end sequenced with a read set-up of 76:6:8:76 using a MiSeq Reagent Kit v3 150 cycles (illumina, MS-102-3001) on an illumina MiSeq instrument. The quality of each sequencing run was assessed using illumina Sequencing Analysis Viewer (illumina version 2.4.7) and all base call files were demultiplexed and converted into FASTQ files using illumina bcl2fastq conversion software v2.20 but the default setting were changed to allow for 0 mismatches. All steps post gDNA extraction to sequencing data generation and data utility were performed at the

Next Generation Sequencing Platform, University of Bern, Switzerland.

For samples from hiPSC-Mac, barcodes were amplified from 100 ng DNA using 31 PCR cycles. For mouse footpad tissue samples, 2000 ng DNA was used as input, with 35 PCR cycles. Amplicons were purified using SPRI magnetic beads and multiplexed by pooling them in equal proportions. The pooled library was once again bead purified and quantified by qPCR using NEBNext Library Quant Kit (NEB, E7630). The library size was verified using a High Sensitivity DNA Kit on a 2100AB Bioanalyzer instrument (Agilent). For whole genome sequencing, a total of 600 ng of genomic DNA from the relevant cell lines was fragmented using DNA fragmentase (NEB, M0348) for 25 minutes at 37 °C. The reaction was stopped using 5 µl of 0.5 M Ethylenediaminetetraacetic Acid (EDTA, Sigma). Fragments of about 350 bp were purified and single-indexed DNA libraries were prepared using standard Illumina TruSeq Nano DNA Library Prep (Reference Guide 15041110 D) and Indexed Sequencing (Overview Guide 15057455). Library fragment size and concentrations were quality verified by qPCR using the NEBNext Library Quant Kit (NEB, E7630) and loading 1:1, 1:3, 1:10 and 1:50 dilutions of the library on a Bioanalyzer High sensitivity DNA Chip (Agilent) and running it on Bioanalyzer 2100AB (Agilent). The amplicon pool was diluted to 4 nM and mixed with 20-40 % single-indexed *Leishmania* genomic DNA. The final library was spiked with 1 % PhiX DNA and the Illumina sequencer was loaded with 8 pM to allow low cluster density ( ~ 800 K/mm²). Sequencing was performed at the Glasgow Polyomics facility using NextSeq Mid Output v2 150 bp kits, with paired-end sequencing (2×75 bp), following the manufactures instructions. NextSeq raw files were de-multiplexed using bcl2fastq (Illumina).

Sequencing reads containing barcode sequences were grouped and counted by scanning sequencing reads for the correct left (GTGTATCGGATGTCAGTTGC) and right (GTATAATGCAGACCTGCTGC) flanking sequences (allowing up to two mismatches per flank) and extracting the 17nt barcode using the python library LeishFASTQ[81]. Previously designed barcode sequences for all genes have been reported in ref. 77, therefore reads were grouped into these four categories:

(i) Invalid reads: sequencing reads that do not contain a valid barcode-like sequence (incorrect or non-existent flanking sequence while allowing for two mismatched nts, barcode sequence of the wrong length).

(ii) Unknown barcode sequences: sequencing reads that contain a valid barcode-like construct, but the barcode sequence does not exist in the LeishGEdit database.

(iii) Foreign barcode sequences: sequencing reads that contain a valid barcode sequence that exists in the LeishGEdit database (allowing a single nt mismatch in the barcode sequence) but the corresponding cell line had not been included in the pool of cell lines.

(iv) Pool member barcode sequences: sequencing reads that contain a valid barcode sequence of a cell line that had been included in the pool (allowing a single nt mismatch in the barcode sequence).

The sequence read counts for each of these categories are shown in Supplementary Data 4 for all sequencing samples.

## Quantification of barcoded cell line fitness

Barcode proportions were calculated by dividing the detected abundance of a given barcode in the sample by the total number of barcoded reads generated from that sample to control for variation in sequencing sample read depths. The barcode change over time was assessed by taking the barcode proportion at a given time point and dividing by the barcode proportion at the 0 h time point for each replicate separately to control for variation in barcoded cell line

prevalence in the population before the start of the assay. Cell line fitness scores were assigned by dividing the median of the corresponding barcode change over all replicates with the median change of all parental cell line barcodes in all replicates. A fitness score above one indicates that proportion of barcodes from a particular cell line have increased relative to the parental cell line reference, corresponding to faster growth and/or better survival than the parental cell line from the start of the assay up to that time point. A fitness score below one indicates the inverse. P-values were calculated using a Mann-Whitney U test against the null hypothesis that the barcode changes from all replicates of a particular cell line in a given time point cannot be distinguished from the barcode changes of all parental cell lines in all replicates of the same time point. Cell lines were labelled as having a strong fitness phenotype in a given time point if their p-value was below 0.05 and their fitness score was either below 0.5 (deleterious phenotype) or above 2 (beneficial phenotype).

### Genomic data processing

Genomic sequencing data were trimmed using Trim Galore version 0.6.6 to remove adaptor sequence and bases where the Phred-scored quality was below 20 from the 3′ end of each read. Reads that were less than 20 bp after trimming were discarded. Trimmed reads were aligned to the *Leishmania mexicana* MHOMGT2001U1103 reference genome obtained from build 55 of TriTrypDB. Alignment was carried out using bowtie2 version 2.4.4[82] using default parameters.

### Confirming knockouts from genomic data

Quantitative coverage data were extracted from the alignments in bigwig format using Deeptools bamCoverage version 3.5.0[83], using a 50 bp bin size and normalised using the Counts Per Million (CPM) method, where the number of counts per bin is divided by the total number of mapped reads in millions. For each knockout, normalised data from the relevant line and from the Cas9 T7 parental line were plotted over the genomic region of the gene that was expected to be knocked out +/- 1.5 kb using the GViz package version 1.40.1[84] in R version 4.2.0[85].

### Estimating ploidy

Quantitative coverage data were extracted from the alignments in bedgraph format using Deeptools bamCoverage[83] using a 1 kb bin size. Assuming that the majority of chromosomes are diploid, scaling to the ploidy was carried out by dividing the coverage for each bin by the median coverage over all bins divided by 2 (coverage/(medianCoverage/2)). The distribution of scaled coverage values was plotted for each chromosome in each line with the Cas9 T7 parental line for comparison, using ggplot2 version 3.3.6[86] in R. Additionally, an estimate of the ploidy of each chromosome in each line was obtained by dividing the median coverage over all the bins in the chromosome by the median coverage over all the bins in the genome divided by 2 (medianCoverageChromosome/(medianCoverageGenome/2)).

### Fluorescence microscopy

Log-phase promastigotes and axenic amastigotes were harvested by centrifugation at 800 $g$ for 5 minutes. Cells were washed twice with PBS and resuspended in PBS containing 20 μM Hoechst 33342. 1 μl of promastigote cell suspension was placed on a poly-L-lysine coated glass slide and imaged immediately. 20 μl axenic amastigote cell suspension was settled on a poly-L-lysine coated glass slide in a humid chamber at 34°C, 5% $CO_2$ for 15 minutes. Excess liquid was discarded before imaging. Cells were imaged on a DM6000 B microscope (Leica Microsystems) with a numerical aperture (NA) 1.30 100x oil immersion objective and Leica DFC360 FX camera. Untagged *L. mex* Cas9 T7 cells were imaged alongside the tagged cell lines with the same exposure time for the GFP channel (Excitation 450-490 nm, Emission 500-550 nm). Micrographs were processed using Fiji[87] and untagged *L. mex* Cas9 T7 were used for background correction of autofluorescence.

### Reporting summary

Further information on research design is available in the Nature Portfolio Reporting Summary linked to this article.

## Data availability

The raw sequencing files (fastq) for all DNA samples from pooled bar-seq screens have been deposited in the the European Nucleotide Archive under accession code: PRJEB76744. The processed bar-seq data are provided in the Supplementary Data 4 file. Whole genome sequences of *L. mexicana* mutants ΔLmxM.03.500, ΔLmxM.08_29.2780, ΔLmxM.08_29.1640, ΔLmxM.22.1420, ΔLmxM.32.3040, ΔLmxM.31.2660 and ΔLmxM.31.1110 have been deposited in the European Nucleotide Archive under accession code PRJEB76744. The whole genome sequence of *L. mexicana* MNYC/BZ/62/M379 C9T7 is available under accession number PRJNA853937[77], Source data are provided with this paper.

## Code availability

The python library LeishFASTQ is at https://zenodo.org/records/14288099[81]. Previously published computational tools were from the following sources: Bowtie2 v2.4.4, http://bowtie-bio.sourceforge.net/bowtie2/index.shtml; Deep tools bamCoverage v3.5.0, https://deeptools.readthedocs.io/en/develop/content/tools/bamCoverage.html; GViz package v1.40.1, https://bioconductor.org/packages/release/bioc/html/Gviz.html; R v4.2.0, https://www.r-project.org/: ggplot2 package v3.3.6, https://cran.r-project.org/web/packages/ggplot2/index.html); Trim Galore v0.6.6 (https://altoslabs.com/ https://github.com/FelixKrueger/TrimGalore), Bcl2fastq (https://support.illumina.com/sequencing/sequencing_software/bcl2fastq-conversion-software.html). Source data are provided with this paper.

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

## Acknowledgements

EG was supported by a Royal Society University Research Fellowship (UF160661). AAW was the recipient of a Marie Skłodowska-Curie Individual Fellowship (trans-LEISHion-EU FP7, No. 798736). TB was supported by MRC PhD studentship (15/16_MSD_836338; https://mrc.ukri.org/), EMBO Postdoctoral Fellowship (ALTF 727-2021) and Marie Skłodowska-Curie Actions Postdoctoral Fellowship (101064428 – LeishMOM). RJW is supported by a Wellcome Trust Henry Dale Fellowship (211075/Z/18/Z). The James and Lillian Martin Centre for Stem Cell Research (SAC) is supported by James Martin 21st Century Research Foundation. This work was supported by a UKRI Medical Research Council grant (MR/V000446/1; This UK funded award is part of the EDCTP2 programme supported by the European Union), the Wellcome Trust (221944/A/20/Z, 200807/Z/16/Z, 104627/Z/14/Z) and the Wellcome Centre for Integrative Parasitology (WCIP) core Wellcome Centre Award (104111/Z/14/Z) and a project grant from the Swiss

National Science Foundation. We like to thank Amanda Williams (University of Oxford), Julie Galbraith and Csilla Balazs (University of Glasgow, Polyomics facility), Pamela Nicholson and Daniela Steiner (NGS Facility, University of Bern) for help with Illumina sequencing, William James (James & Lillian Martin Centre, Sir William Dunn School of Pathology, University of Oxford) for supporting the work with iPSC-derived macrophages, Caroline Ricce Espada for generating the Ros3 knockout cell line, Keith Gull (Sir William Dunn School of Pathology, University of Oxford) for access to equipment and all past and current members of EG lab for helpful discussions.

## Author contributions

Conceptualisation, A.A.W., C.M.C. and E.G. Methodology, T.B., U.D., R.J.W. and E.G. Software, U.D. and R.J.W. Formal analysis, A.A.W., U.D. and E.G. Investigation, A.A.W., C.M.C., R.N., C.A., T.B. and K.C. Resources, E.G., J.C.M. and S.A.C. Data curation, A.A.W., E.G., U.D. and R.J.W. Writing original draft, A.A.W. and E.G. Writing review and editing, A.A.W., C.A., R.J.W., T.B., R.N., J.C.M., E.G. Visualisation, A.A.W. and C.A. Supervision, E.G. and A.A.W. Project administration, E.G. Funding Acquisition, E.G., A.A.W., R.J.W. and J.C.M.

## Competing interests

The authors declare no competing interests.
