## [Transparent Peer Review file · Nature Communications]

TransLeish: Identification of membrane transporters essential for survival of intracellular *Leishmania* parasites in a systematic gene deletion screen

Corresponding Author: Professor Eva Gluenz

Version 0:

Reviewer comments:

Reviewer #1

(Remarks to the Author)

The authors conduct a large study, first to identify the list of genes that are most likely to be transporters in *Leishmania mexicana*. Then using CRISPR/Cas9 they endeavour to make mutants for each of these candidates, and then quantify their role in fitness both in promastigote and amastigote stages (macrophage and mouse infections). This is a huge effort and will be beneficial to the field. The experiments are sound, logically done, and have interesting results. However, there is no major singular insight into *Leishmania* biology as a result of this study, this work is more of an initial step to help prioritise follow-up experimentation. That alongside some other issues with how the data is explained and presented make me unsure that this work is the right fit for Nature Communications at this time.

First in the introduction, the authors made a strong case about the role of transporters in communicating between the host and the parasite, and how transporters at the plasma membrane that directly interfaced with the PV would be critical for providing nutrients and maintaining homeostasis for the parasite. However, later in the manuscript, several of the transporters involved in the study are hypothesised to be localised to organelles (for example mitochondrial carrier proteins). From the introduction, I thought the "transportome" being studied was the transporters at the interface with the host, not all transporters expressed in *Leishmania mexicana*. I found this misleading.

I also was a bit surprised to read that the main way to generate the list of transporter candidates was a keyword search. This feels very underpowered bioinformatically. It is also presented a little too strongly as a final list, where I think it is more of a candidate list of transporters.

I didn't find it that surprising that different transporters would be required in promastigote vs amastigote, as the culture/host conditions are very different and the life cycle stages will have different metabolic needs. I found the description of the potential function of the V ATPase interesting, but the authors did not perform much follow-up experimentation. In the result section line 374-376 they suggest that this complex transporter is important for amastigotes to survive the acidic environment of the PV, and that is why they didn't recover the null mutants in macrophage/mice pooled experiments, however there is no direct evidence reporter in the manuscript.

Overall I also found the figures a bit challenging. They could be presented more clearly with a number of minor edits to make them easier to understand.

While this manuscript represents a significant amount of work, it mostly reports the outcome of a fitness screen for transporters in promastigote vs amastigote stage. I feel that more follow-up describing in-depth detail of the function of a few of these transporters in their role in parasite biology is required.

Issues with figures:

-All Figures: I am not sure that the color scheme selected is color-blind friendly. I also find it difficult to distinguish between the more subtle differences in shades of green. I think this should be reconsidered on the whole. Figure legends lack the number of biological repeats, technical replicates, statistical tests, and what is being plotted (mean, median, error bars, etc?).

There is not enough information in the figure legend. Sometimes the lettered panels are in columns and sometimes in rows- this could be harmonised and it would be a lot easier to read. Also I think rows would be better, I have not seen so many figures where panels are arranged in columns.

-Figure 1: I am not sure how much this adds to the manuscript. If you are trying to draw attention to the number of transporters in *L. mexicana*, this sample should be highlighted in some way so that it's clear where it falls amongst the context of other organisms. Species names underneath should be a smaller font and at an angle to make it easier to read. The symbols in the key are too small to see the assigned color easily. It is not clear to me how the numbers were arrived at. For example the number of predicted transporters for *Cryptosporidium* appears too low to me (PMID: 37138353 says there are 152 transporters).

-Figure 2: Panel B and C have their own color scheme, which slightly overlaps the colors used in A, but doesn't actually correspond to the colors used in A. I find this very confusing as I was looking at the panel as a whole. I also think that it is very difficult to look at 49 pie charts and make much sense of things when there are so many acronyms and numbers. Of course this is the nature of the work, but I don't think a series of pie charts is the best way to convey the complexity of the data. Also the size of the circles changes for each row, and it is not clear why and what this is meant to convey. I would reconsider how this figure is illustrated.

-Figure 3: Panel B I find the dots to be too big (they obstruct the line in the background in this panel and all similar panels containing a fitness curve) and the gray color gets washed out. It is not clear in C why these collection of genes were "selected" and the use of asterisks to both indicate incomplete mutants and statistical significance is confusing.

-Figure 4: Again, dots are too big and make it difficult to really see where they points lie on the curve.

-Figure 7: Macrophage and mouse experiments should be more clearly labeled.

Reviewer #2

(Remarks to the Author)

This manuscript describes a comprehensive in vitro and in vivo analysis of the roles of membrane transporters in the *Leishmania* life cycle and pathology. As such it provides a wealth of new insight which will impact upon our understanding.

The level of experimental detail is admirable.

The work is well conducted and results clearly presented - of 312 putative transporters, all identified by literature and database searches, 188 nulls and 81 partial nulls were isolated. Good to see WGS employed for verification of some cell lines. The gene array deletion work is also impressive (two arrays successfully deleted) – giving confidence in the data as a whole.

Proton pump genes appear particularly important in infection – maybe not surprisingly.

Important to note that only nine viable nulls of single member superfamily KOs were generated alongside these two array KOs. So, 11 in total. However, analyses proceeded with all 251 viable mutants obtained. More specific detail on the outcomes for these 11 would be of use.

SPECIFIC POINTS - MAJOR ONES HIGHLIGHTED

Introduction

56-72 – lightly referenced, maybe add some citation to reviews etc.

73-74 – are membrane transporters also likely to important in insect stages?

88 – “...other trace elements”? Any evidence for what, citations?

Results

MAJOR, A triage figure on arriving at the 251 viable mutants – and which are the 188 KOs – would be a very good addition.

154 – MAJOR, Figure 1, data would be better presented as percentage of genes.

160-161 – “This identified 49 different superfamilies for which *L. mexicana* has 161 between one and 53 protein members.” I assume these identifications are largely putative?

195-199 – MAJOR, Supplementary Figure 3, further comment on and consideration of ploidy changes is warranted. This could have a major effect on the phenotype.

286 – “61 knock-outs of non-transporter genes” – what are these, where did they come from?

300-301 – “The fitness scores of the parasite lines tested in mice spanned a larger range compared to the lines tested in macrophage infections in vitro.” Could this be due to the very different ‘incubation’ times?

Figure 4B-H – the x-axis could be altered (elongated?) to make data clearer.

342-343 – The focus on the V-ATPase pump is justified – but at this point I wondered what of the 11 ‘complete KOs’ – those either with no other superfamily member or with the array KO’ed out?

Reviewer #3

(Remarks to the Author)

Membrane transport proteins play critical roles for parasites, which must salvage a variety of nutrients and micronutrients from their hosts, often in competition with host transporters. While previous work has addressed the role of various transporters on a piecemeal basis, this study is the first to define the importance of the entire transportome for *Leishmania* parasites, especially for the intracellular mammalian infectious stage that causes disease. Confirmed gene knockouts in promastigotes have been obtained for 188 (60%) of the predicted 312 transporter genes in *L. mexicana*, and ‘incomplete’ knockouts have been isolated for another 81 genes. Most of the knockouts are new, thus providing a resource for biological characterization of these transporters in parasite physiology even though they are not essential permeases. This work represents a significant advance in providing a synoptic view of the role of transporters in parasite fitness.

Incomplete knockouts obtained in promastigotes, in which a copy of the targeted gene was retained despite integration of two drug resistance cassettes plus 11 transporter genes that could not be deleted following two attempts, suggest that many of those transporters may play critical roles in the physiology of the parasites, potentially in both life cycle stages. In addition, among the confirmed null mutants, 13 adversely affected growth rate of promastigotes. Overall, these results provide valuable knowledge regarding which transporters are likely to be critical for promastigotes, although as the authors point out, more work on individual genes will be required to determine which candidates are truly important for parasite fitness and which knockout failures may represent technical problems. Importantly, this study also investigated the role of 254 transporter mutants, both confirmed and incomplete knockouts, in both macrophages in vitro and in infected mice, thus addressing the roles of transporters in the disease-causing stage of the parasite life cycle. 17 mutants showed reduced fitness in both *Mφ* and mice and another 11 showed reduced fitness only in mice. Hence, this study has identified a cohort of transporters that play critical roles in intracellular disease-causing amastigotes, and these results are of high significance for the field. Notably, 10 of these null mutants were for subunits of the vacuolar H⁺-ATPase and one was for a P-type H⁺-ATPase, underscoring the importance of controlling intracellular pH for the amastigotes, which live in the acidified parasitophorous vacuole of host macrophages. These proteins may represent valuable targets for development of anti-leishmanial drugs, further supporting the significance of this study.

Specific Comments.

1. A major conclusion of this manuscript is that the multi-subunit V H⁺-ATPase is critical for viability of intracellular amastigotes that live in an acidic environment, apparently by maintaining intracellular pH homeostasis. It would enhance the significance and impact of this paper if the authors could determine the localization of this critical V H⁺-ATPase in the amastigotes, as suggested in lines 472-474 and 503-504; this would help clarify its mode of action as a determinant of pH homeostasis. Is it partially a surface proton pump, or does it function exclusively in intracellular organelles? Tagging one or two subunits and determining localization by fluorescence microscopy would be a relatively straightforward experiment.
2. In Fig. 5A, B, some incomplete mutants (marked by *) had strong loss of fitness as amastigotes. Do the authors think that these phenotypes are due to dosage-dependent loss of fitness, since the only probable difference compared to wild type parasites is the reduced copy number of the target gene?
3. Lines 330-337. The authors should explain why the data suggest a conditional loss of fitness for the indicated null mutants.
4. Around line 430, it may be worth noting that some of the confirmed null mutants that are viable as promastigotes in culture could nonetheless have significant fitness defects under natural conditions in the sand fly.
5. In lines 652-658, the authors indicate that they assigned the transporter mutants to pools of normal, slow, and very slow growth. They should indicate in the Methods section how they measured these growth rates for such binning. It seems that the section on Parasite Growth Curves in line 561 refers only to growth under different pH conditions.
6. Instead of referring to CAKC as a ‘calcium/potassium channel’ (e.g., line 334 and elsewhere), the authors should call this a calcium-activated potassium channel.
7. The legend for Supple Fig. 4C is confusing and would benefit from rewriting.
8. Data is a plural noun and should always be followed by a plural verb. There are several places (e.g., line 331) in the manuscript where singular verb forms are used.
9. Line 70 should be modified to indicate ‘the acid tolerant amastigote form’.

Version 1:

Reviewer comments:

Reviewer #1

(Remarks to the Author)

Thank you for clarifying in the introduction that the list of transporters includes both surface-localised transporters as well as organellar transporters.

Thanks also for clarifying the methods used to generate the list of candidate transporters. This really improves the understanding of how this list was created.

The figures have been greatly updated along with the legends, and this had made a significant impact in improving the manuscript.

The localisation and updated figures have streamlined the narrative of the outcome of this screen. This data will be very useful to the wider parasitology field and is an impressive amount of work.

The comments from the other reviewers are also well addressed.

Reviewer #2

(Remarks to the Author)

All my points addressed - thank you.

Reviewer #3

(Remarks to the Author)

I have looked over the revised manuscript and the responses to all three reviewers' comments. I am satisfied that the authors have answered appropriately all questions, with one reservation below, and I support progressing with the revised manuscript.

I have one comment that I believe the authors should address, ideally experimentally but at least in their Discussion section. As requested, the authors have added a new experiment localizing two subunits of the V-ATPase that plays a critical role in amastigote viability. The images in Fig. 7E show a localization to a structure that they interpret as endosomal vesicles. The endosomes do emerge from the flagellar pocket membrane, so given the location they observe that suggestion is reasonable. However, without a marker, it is difficult to distinguish such vesicles from the flagellar pocket itself, which is indeed located close to the kinetoplast that can be seen in purple nearby the green fluorescence in the images. Hence, it is possible that V-ATPase is located, partially or completely, in the flagellar pocket membrane. In this case, a plausible explanation for its function could be that it pumps cytosolic protons into the flagellar pocket, thus maintaining internal pH homeostasis in the amastigote.

made.

We thank all three reviewers for their detailed and helpful comments on the manuscript. We are pleased with the positive reception of the work. In our revision, we incorporated suggestions for improving the presentational clarity of the data and we provided the requested additional information on the subcellular localization of the V-ATPase. The reviewers' comments have been very helpful in guiding the revision of the manuscript and we provide detailed responses to each comment below.

REVIEWER COMMENTS FOR NCOMMS-24-41748

Reviewer #1 (Remarks to the Author):

The authors conduct a large study, first to identify the list of genes that are most likely to be transporters in *Leishmania mexicana*. Then using CRISPR/Cas9 they endeavour to make mutants for each of these candidates, and then quantify their role in fitness both in promastigote and amastigote stages (macrophage and mouse infections). This is a huge effort and will be beneficial to the field. The experiments are sound, logically done, and have interesting results. However, there is no major singular insight into *Leishmania* biology as a result of this study, this work is more of an initial step to help prioritise follow-up experimentation. That alongside some other issues with how the data is explained and presented make me unsure that this work is the right fit for Nature Communications at this time.

First in the introduction, the authors made a strong case about the role of transporters in communicating between the host and the parasite, and how transporters at the plasma membrane that directly interfaced with the PV would be critical for providing nutrients and maintaining homeostasis for the parasite. However, later in the manuscript, several of the transporters involved in the study are hypothesised to be localised to organelles (for example mitochondrial carrier proteins). From the introduction, I thought the "transportome" being studied was the transporters at the interface with the host, not all transporters expressed in *Leishmania mexicana*. I found this misleading.

Response from the authors:

> In the introduction we outline broadly the fundamental importance of transporters as gatekeepers on the cell surface and on internal membranes and emphasize their varied roles in cellular physiology. Transport of nutrients is one of the key roles, and particularly interesting in the context of host-parasite interactions, but it is not the only one, and nutrient uptake and cellular physiology are interlinked. We have added two lines to the introduction to make clear that adaptation to the intracellular niche requires the amastigotes both to scavenge nutrients and to adapt their cellular physiology to the new environment, and we have shortened the paragraph on the metabolic switch in amastigotes. Our global mutant screen of the "transportome" is comprehensive (surface and organellar – we have also added a line to text in the introduction to clarify this point early) including all transporter-encoding genes we found in the genome of *L. mexicana*, only excluding the specific categories mentioned in the text.

I also was a bit surprised to read that the main way to generate the list of transporter candidates was a keyword search. This feels very underpowered bioinformatically. It is also presented a little too strongly as a final list, where I think it is more of a candidate list of transporters.

>We have now expanded the methods section such that the strength of the underlying bioinformatics becomes clearer to the readers. To summarise the key points:
Our list of transporter protein encoding genes in the genome of *L. mexicana* is the result of an extensive iterative search strategy combining integration of results from bioinformatics and deep reading of the literature.

A keyword search served as our entry point. It pulled together already curated information on *L. mexicana* proteins (linked to information about all other kinetoplastids represented in TriTrypDB) and *Leishmania* transporters (TransportDB 2.0). This allowed us to gather information on relevant sequence features, including Pfam domains, which are defined from profile hidden Markov models (HMM), Prosite patterns, which are based on multiple sequence alignments and Prosite profiles (weight matrices) and other signatures that are computed by the member databases of InterPro. These signatures were then used as starting points for finding additional candidate sequences. Curation of the list then involved BLAST searches and assessment of multiple sequence alignments using the curated transporter classification database TCDB, which is an extensive collection of transporter proteins from prokaryotes and eukaryotes.

We used two different methods to predict transmembrane domains and we extensively curated our own list of candidate proteins by cross-checking sequences against the published literature to clarify for example the identity of transporters described in older literature before GeneIDs were available. In summary, our list of predicted transporter proteins in *L. mexicana* is the result of a comprehensive and multifaceted search strategy, underpinned by solid bioinformatic evidence. It is the most comprehensive curated "candidate" list for any kinetoplastids species that we are aware of. We trust the extended description of the methods clarifies the scope of the analysis.

I didn't find it that surprising that different transporters would be required in promastigote vs amastigote, as the culture/host conditions are very different and the life cycle stages will have different metabolic needs. I found the description of the potential function of the V ATPase interesting, but the authors did not perform much follow-up experimentation. In the result section line 374-376 they suggest that this complex transporter is important for amastigotes to survive the acidic environment of the PV, and that is why they didn't recover the null mutants in macrophage/mice pooled experiments, however there is no direct evidence reporter in the manuscript.

>The underlying hypothesis for this study was that promastigotes and amastigotes require different sets of transporters (with some overlap) and that these requirements could be revealed by testing the fitness of knockout mutants in the respective environments. The identity of the transporters that are vital for amastigotes is of interest to understand the basic cell biology of these forms, and to identify possible points of attack for the development of anti-parasitic compounds. Which of the >300 transporter proteins are vital for the amastigotes had never been tested in a systematic study. We reasoned, that a knockout screen would allow us to generate viable promastigote form null mutants for transporters that were essential only in the amastigote form; their subsequent exposure to the conditions in macrophages and mice would reveal which of these mutations were detrimental, thus identifying the transporters of interest. The study achieved this aim and as a result we now have for the first time quantitative fitness measures for 188 transporter KO mutants in promastigotes and amastigotes.

>Like the reviewer, we also thought the The V-ATPase phenotype was interesting. This highly conserved complex likely has a number of different important functions in the cell. Studying these in detail merit further detailed investigations which are beyond the scope of this study. From this screen we can draw two main conclusions about the V-ATPase.

1. This complex is vital for *Leishmania* in mice and macrophages but dispensable in promastigotes under standard laboratory culture conditions.
2. When promastigotes are subjected to acidic conditions, the V-ATPase mutants proliferate poorly, indicating that *Leishmania* require V-ATPase in acidic environments.

>Acidity is one of the hallmarks of the intracellular environment in which amastigotes thrive, therefore we consider the reduced pH a possible reason why V-ATPase mutants drop out from the mutant pools in mice and macrophages. Given that the V-ATPase likely supports a multitude of cellular processes, the possibility remains open that the loss of V-ATPase has other, perhaps additive detrimental effects on cellular physiology that impact on the amastigotes, in addition to rendering the *Leishmania* more pH-sensitive.

Overall I also found the figures a bit challenging. They could be presented more clearly with a number of minor edits to make them easier to understand.

>A number of edits have been made to the figures, helped by suggestions from all three referees.

While this manuscript represents a significant amount of work, it mostly reports the outcome of a fitness screen for transporters in promastigote vs amastigote stage. I feel that more follow-up describing in-depth detail of the function of a few of these transporters in their role in parasite biology is required.

>We believe the function of transporters in *Leishmania* is an important and relatively understudied area, which would benefit from detailed studies into the substrate specificity of individual transporters, their transport mechanisms at a molecular/structural level, their functions in cellular physiology, redundancies among members of transporter families, their contributions to host-parasite cross-talk and competition and their contributions to drug uptake and resistance, to name but a few. Here we present the first "transportome"-wide fitness screen for *Leishmania*, providing for 188 transporters a measure of their relative importance in promastigotes vs. amastigotes. We

believe this comprehensive study will help prioritise transporters for detailed follow-up investigations, which merit their own dedicated studies.

Issues with figures:

-All Figures: I am not sure that the color scheme selected is color-blind friendly. I also find it difficult to distinguish between the more subtle differences in shades of green. I think this should be reconsidered on the whole. Figure legends lack the number of biological repeats, technical replicates, statistical tests, and what is being plotted (mean, median, error bars, etc?). There is not enough information in the figure legend. Sometimes the lettered panels are in columns and sometimes in rows- this could be harmonised and it would be a lot easier to read. Also I think rows would be better, I have not seen so many figures where panels are arranged in columns.

> After revising our original figures, across all colour-blind impairments, we indeed identified that people with the rare type of red-blind/protanopia, would likely find it difficult to distinguish the colours chosen for the original submission. In response, we have revised the colour palette of all figures, using a colour palette designed for better accessibility, as recommended by Katsnelson A. Colour me better: fixing figures for colour blindness. *Nature*. 2021 Oct;598(7879):224-225. We individually checked each final version of altered figures using the Coblis colour blindness simulator (<https://www.color-blindness.com/>), and confirmed sufficient contrast is provided for data interpretation. This adjustment ensures that the figures are now more inclusive and can be interpreted accurately by most readers. The authors appreciate this reviewer's attention to this important matter and the opportunity to significantly improve the clarity and accessibility of this manuscript.

>We have re-arranged Figure panels in rows, rather than columns.

>Figure legends have been amended to include more details.

-Figure 1: I am not sure how much this adds to the manuscript. If you are trying to draw attention to the number of transporters in *L. mexicana*, this sample should be highlighted in some way so that it's clear where it falls amongst the context of other organisms. Species names underneath should be a smaller font and at an angle to make it easier to read. The symbols in the key are too small to see the assigned color easily. It is not clear to me how the numbers were arrived at. For example the number of predicted transporters for *Cryptosporidium* appears too low to me (PMID: 37138353 says there are 152 transporters).

>We have enhanced the presentation of this figure according to the reviewers' helpful suggestions. We believe this figure is helpful to the readers as it places our predicted *L. mexicana* transportome in the context of estimated transporter numbers for other species.

-Figure 2: Panel B and C have their own color scheme, which slightly overlaps the colors used in A, but doesn't actually correspond to the colors used in A. I find this very confusing as I was looking at the panel as a whole. I also think that it is very difficult to look at 49 pie charts and make much sense of things when there are so many acronyms and numbers. Of course this is the nature of the work, but I don't think a series of pie charts is the best way to convey the complexity of the data. Also the size of the circles changes for each row, and it is not clear why and what this is meant to convey. I would reconsider how this figure is illustrated.

>We have now unified the colour scheme. With the pie charts in panel B we wish to show two main pieces of information: 1. A visual overview of the transporter family repertoire in *Leishmania* and the relative size of the families (the number of family members is written in the circles and the different sizes are a qualitative representation of family size, with bigger circles indicating more populated families – we have amended the figure legend to explain this more clearly). 2. A visual overview of the KO status for each family, allowing for easy identification for example of superfamilies that were completely dispensable in cultured promastigotes (discussed in the results).

-Figure 3: Panel B I find the dots to be too big (they obstruct the line in the background in this panel and all similar panels containing a fitness curve) and the gray color gets washed out. It is not clear in C why these collection of genes were "selected" and the use of asterisks to both indicate incomplete mutants and statistical significance is confusing.

>We have revised the presentation of Figures 3 and 4 and made the coloured dots semi-transparent for greater clarity. The dual use of asterisks was an oversight, which we have now corrected, using

a different symbol to denote the incomplete knockouts. The mutants were chosen because they were observed to grow slowly during the selection process. These measurements served to quantify these observations and to allow for comparisons of fitness effects measured in pooled screens with individually measured growth rates.

-Figure 4: Again, dots are too big and make it difficult to really see where they points lie on the curve.

>See above.

-Figure 7: Macrophage and mouse experiments should be more clearly labeled.

>Each sub-panel is now labelled with the condition and timepoint.

Reviewer #2 (Remarks to the Author):

This manuscript describes a comprehensive in vitro and in vivo analysis of the roles of membrane transporters in the *Leishmania* life cycle and pathology. As such it provides a wealth of new insight which will impact upon our understanding.

The level of experimental detail is admirable.

The work is well conducted and results clearly presented - of 312 putative transporters, all identified by literature and database searches, 188 nulls and 81 partial nulls were isolated. Good to see WGS employed for verification of some cell lines. The gene array deletion work is also impressive (two arrays successfully deleted) – giving confidence in the data as a whole.

Proton pump genes appear particularly important in infection – maybe not surprisingly.

Important to note that only nine viable nulls of single member superfamily KOs were generated alongside these two array KOs. So, 11 in total. However, analyses proceeded with all 251 viable mutants obtained. More specific detail on the outcomes for these 11 would be of use.

>The fitness data for the single-member family KOs are in Supplementary Table 4. The only one of these that showed a significant loss-of-fitness phenotype was the KO for the vacuolar iron transporter (VIT family; LmxM.21.1580), which was significantly depleted after three and six weeks in the mouse (shown in Figure 5). Although the importance of iron homeostasis on amastigote forms is well documented, and VIT transporters have been studied in other protozoan parasites, we could not find any previous reports of the VIT transporter being studied in *Leishmania*. Therefore we have now added a sentence to the discussion to highlight this finding.

SPECIFIC POINTS - MAJOR ONES HIGHLIGHTED

Introduction

56-72 – lightly referenced, maybe add some citation to reviews etc.

>We have corrected this omission and added relevant references to the introductory paragraph.

73-74 – are membrane transporters also likely to important in insect stages?

> As noted at the beginning of the introduction, controlled transport across membranes is fundamental to cellular physiology and the maintenance of homeostasis, this applies to all forms of *Leishmania*, in all of the environments. In this study we examine the fate of transporter mutants in macrophages and in mice. We also assess mutant fitness in cultured promastigotes, which could be called “insect stage”, but we presume the reviewer means promastigotes in their natural sand fly vector, which we did not study here. Although this is clearly a very important and interesting question in itself, we refrained from elaborating on the conditions in the sand fly and how they may differ from those in laboratory cultures, in order to keep the introduction concise. We have amended the discussion to say that some of the transporters that were non-essential for cultured promastigotes may prove to be vital for passage through a sand fly.

88 – “.other trace elements”? Any evidence for what, citations?

>We have now specified examples of trace minerals that were shown to support *Leishmania* growth, with reference to the paper by Nayak et al., 2018.

Results

MAJOR, A triage figure on arriving at the 251 viable mutants – and which are the 188 KOs – would be a very good addition.

>Rather than attempting to list the mutants in a figure, we provide the full list of all targeted genes and the genotype of each mutant in Supplementary Table 1. We have now added this reference to the relevant line in the results to guide the reader to the data.

154 – MAJOR, Figure 1, data would be better presented as percentage of genes.

>The figure shows what percentage of proteomes is comprised of transporter proteins. Both values are mostly predictions, based on genome annotations. The size of the proteomes was taken from UniProtKB, which is a combination of curated Swiss-Prot data and translated genome sequences (TrEMBL). We chose to take this transparent and accessible measure, to avoid having to duplicate efforts to define a total number of genes for each species, or determine which genes are likely to be protein-coding. In our opinion the UniProtKB data this is one of the best available estimates of proteome size. Likewise, the numbers for predicted transporters were taken from TransportDB, a central repository for transporter sequence data. We have amended the figure legend to make it clearer what the data sources are to allow readers to replicate the findings and build on these data.

160-161 – “This identified 49 different superfamilies for which *L. mexicana* has 161 between one and 53 protein members.”

I assume these identifications are largely putative?

>The identifications are based on BLAST searches of TCDB and it is therefore correct to say these are putative identifications. We modified the text to make this explicit.

195-199 – MAJOR, Supplementary Figure 3, further comment on and consideration of ploidy changes is warranted. This could have a major effect on the phenotype.

>We assessed seven mutants in detail by whole genome sequencing. The most noticeable outcome of the ploidy analysis was that chromosome numbers remain largely constant between the parental cell line and the seven different mutants. No changes were observed for any of the chromosomes that contained the targeted gene. We cannot exclude that individual knockouts may lead to compensatory alteration of gene copy numbers elsewhere. This is a caveat of any reverse genetic study. We have added additional information to the text to give two specific instances of ploidy changes of other chromosomes seen in mutants. We say in the paper “whether these changes in chromosome ploidy were stochastic events or causally linked to the specific gene deletions cannot be inferred from these data.” We do not wish to speculate further but would like to support calls for more dedicated studies on the causes and effects of ploidy changes using systematic well-controlled experiments, which would be of great benefit to the *Leishmania* field.

286 – “61 knock-outs of non-transporter genes” – what are these, where did they come from?

> The mutant pools that were passaged through macrophages and mice were combined pools of the TransLeish transporter mutants (and non-transporter controls, as detailed throughout the manuscript) plus another 57 knockout mutants of flagellar proteins that were analysed in a separate study. The pools were combined in order to reduce the number of animals subjected to *in vivo* experiments, in line with 3R principles (<https://nc3rs.org.uk/>). We have now amended the line cited by the reviewer by referring to the methods section, where the pool composition was explained in more detail. In the methods section, we further extended the description of the datasets that are made available with this publication: The bar-seq data deposited in the European Nucleotide Archive ENA under accession PRJEB76744 and Supplemental Table 4 (Tab “Raw Read Counts TLcomb *in vivo*”) contain the bar-seq data and read counts for all cell lines included in the pools. These data

were used to calculate the fitness scores for the transporter mutants and controls that were analysed in detail in the current publication.

300-301 – “The fitness scores of the parasite lines tested in mice spanned a larger range compared to the lines tested in macrophage infections *in vitro*.” Could this be due to the very different ‘incubation’ times?

>It is very likely that the different observation times explain many of the differences between the macrophage infections and the mouse infections relating to the persistence of mutants. Length of time is however not the only factor; the mouse is a much more complex environment where systemic immune responses act on the macrophages. We describe here the outcome of the experiment, with respect to mutant fitness under the tested conditions. From these data alone we can make no inferences about the differences between macrophages *in vivo* and those grown *in vitro*, which may impact differently on *Leishmania* transporter mutants and we do not wish to speculate too much in the discussion. For technical reasons it is not possible to maintain infected macrophages in culture for more than ~1 week, a limitation that underlines the importance of *in vivo* studies. The mouse model is the physiologically more relevant model. The value of adding the macrophage model is that we can use it to identify the mutants that appear to struggle during the initial infection. These include the V-ATPase mutants, which we have followed up.

Figure 4B-H – the x-axis could be altered (elongated?) to make data clearer.

>We have altered the figures for better clarity.

342-343 – The focus on the V-ATPase pump is justified – but at this point I wondered what of the 11 ‘complete KO’s’ – those either with no other superfamily member or with the array KO’ed out?

>Please see the reply above. Of the KOs that removed all members of a superfamily, the vacuolar iron transporter knockout was severely impacted in the mouse (at the 3 and 6-week timepoints, Figure 5). We agree this was worth highlighting and added this information to the discussion.

Reviewer #3 (Remarks to the Author):

Membrane transport proteins play critical roles for parasites, which must salvage a variety of nutrients and micronutrients from their hosts, often in competition with host transporters. While previous work has addressed the role of various transporters on a piecemeal basis, this study is the first to define the importance of the entire transportome for *Leishmania* parasites, especially for the intracellular mammalian infectious stage that causes disease. Confirmed gene knockouts in promastigotes have been obtained for 188 (60%) of the predicted 312 transporter genes in *L. mexicana*, and ‘incomplete’ knockouts have been isolated for another 81 genes. Most of the knockouts are new, thus providing a resource for biological characterization of these transporters in parasite physiology even though they are not essential permeases. This work represents a significant advance in providing a synoptic view of the role of transporters in parasite fitness.

Incomplete knockouts obtained in promastigotes, in which a copy of the targeted gene was retained despite integration of two drug resistance cassettes plus 11 transporter genes that could not be deleted following two attempts, suggest that many of those transporters may play critical roles in the physiology of the parasites, potentially in both life cycle stages. In addition, among the confirmed null mutants, 13 adversely affected growth rate of promastigotes. Overall, these results provide valuable knowledge regarding which transporters are likely to be critical for promastigotes, although as the authors point out, more work on individual genes will be required to determine which candidates are truly important for parasite fitness and which knockout failures may represent technical problems. Importantly, this study also investigated the role of 254 transporter mutants, both confirmed and incomplete knockouts, in both macrophages *in vitro* and in infected mice, thus addressing the roles of transporters in the disease-causing stage of the parasite life cycle. 17 mutants showed reduced fitness in both MØ and mice and another 11 showed reduced fitness only in mice. Hence, this study has identified a cohort of transporters that play critical roles in intracellular disease-causing amastigotes, and these results are of high significance for the field. Notably, 10 of these null mutants were for subunits of the vacuolar H⁺-ATPase and one was for a P-type H⁺-ATPase, underscoring the importance of controlling intracellular pH for the amastigotes, which live in the acidified parasitophorous vacuole of host macrophages. These proteins may represent

valuable targets for development of anti-leishmanial drugs, further supporting the significance of this study.

Specific Comments.

1. A major conclusion of this manuscript is that the multi-subunit V H⁺-ATPase is critical for viability of intracellular amastigotes that live in an acidic environment, apparently by maintaining intracellular pH homeostasis. It would enhance the significance and impact of this paper if the authors could determine the localization of this critical V H⁺-ATPase in the amastigotes, as suggested in lines 472-474 and 503-504; this would help clarify its mode of action as a determinant of pH homeostasis. Is it partially a surface proton pump, or does it function exclusively in intracellular organelles? Tagging one or two subunits and determining localization by fluorescence microscopy would be a relatively straightforward experiment.

>We have addressed this important question and include a new Figure (7E) showing the localization of tagged V-ATPase subunits G and H. The major conclusions are: The localization pattern is the same for both subunits. Importantly the pattern is also the same for promastigotes and amastigotes: The strongest signal is consistently seen adjacent to the flagellar pocket (one might annotate it as "endocytic organelles"), with weaker foci of variable intensities seen throughout the cell body (consistent with a function in several cellular organelles); we did not observe any signal at the plasma membrane. We also consulted relevant data already in the public domain, namely TrypTag and LeishTag, and LOPIT datasets from *T. brucei* (accessed via TritypDB) and *L. mexicana* (LeishGEM.org) which report localisations for V-ATPase subunits that are consistent with a large fraction of V-ATPase proteins being associated with endocytic organelles. We have added relevant references to the text. In short: we do not find any evidence pointing to V-ATPase acting as a surface proton pump in *Leishmania*.

2. In Fig. 5A, B, some incomplete mutants (marked by *) had strong loss of fitness as amastigotes. Do the authors think that these phenotypes are due to dosage-dependent loss of fitness, since the only probable difference compared to wild type parasites is the reduced copy number of the target gene?

>Dosage-dependent loss of fitness could indeed be an explanation, but we do not have an accurate measure of gene dosage for these lines. Mutants are defined as "Incomplete" when both donor DNA cassettes are integrated in the genome, but a diagnostic PCR still detects the target gene. In this screen we cannot distinguish between cases where all of the cells have retained a copy of the gene (heterozygous – perhaps they had more than two copies to start with), or only a small minority of cells retained the gene (resulting in a mixed population of null mutants and heterozygotes).

3. Lines 330-337. The authors should explain why the data suggest a conditional loss of fitness for the indicated null mutants.

>By conditional essentiality we mean genes that are dispensable in one condition but their loss has a severely detrimental effect in another condition. In our screen these are the genes where promastigotes knockouts have similar fitness to the parental line, but amastigotes showed a significant and severe loss of fitness, defined in the text as a fitness score below 0.5 ($p < 0.05$).

4. Around line 430, it may be worth noting that some of the confirmed null mutants that are viable as promastigotes in culture could nonetheless have significant fitness defects under natural conditions in the sand fly.

>This is of course correct and we have amended the text to state this explicitly.

5. In lines 652-658, the authors indicate that they assigned the transporter mutants to pools of normal, slow, and very slow growth. They should indicate in the Methods section how they measured these growth rates for such binning. It seems that the section on Parasite Growth Curves in line 561 refers only to growth under different pH conditions.

>We have now corrected this omission and amended the methods description, which now details both the measurement of growth in standard culture (data for Figure 3) and in cultures with different pH (Figure 7). The categories "normal", "slow", and "very slow" growth were qualitative assessments of cell growth during selection. We have re-phrased the relevant line to state this more clearly.

6. Instead of referring to CAKC as a 'calcium/potassium channel' (e.g., line 334 and elsewhere), the authors should call this a calcium-activated potassium channel.

>We thank the reviewer for this clarification and we have corrected the text where appropriate.

7. The legend for Supple Fig. 4C is confusing and would benefit from rewriting.

>The revised version should provide clarity.

8. Data is a plural noun and should always be followed by a plural verb. There are several places (e.g., line 331) in the manuscript where singular verb forms are used.

>We have corrected these.

9. Line 70 should be modified to indicate 'the acid tolerant amastigote form'.

>We have corrected this.

In addition to the above revisions, we corrected the following errors:

Fig.2 Ca-CIC – the numbers of complete/incomplete KOs was incorrectly given as 2/0. The corrected values are 1/1 as shown in the revised figure and the spelling of the family name has been corrected.

Fig.6B – The GeneIDs for subunits V1 C and Vo c had inadvertently been mixed up in the previous versions of Figures 5 and 6 and Table S1. We have corrected the this in the revised paper to show LmxM.18.0560 is V1 C is and LmxM.21.1800 is Vo c.

We have also added previously missing oligonucleotide sequences to Supplementary Table 5.

None of these corrections alter any of our conclusions.

We thank all three reviewers again for their detailed and helpful comments on the manuscript.

REVIEWER COMMENTS FOR NCOMMS-24-41748A

REVIEWERS' COMMENTS

Reviewer #1 (Remarks to the Author):

Thank you for clarifying in the introduction that the list of transporters includes both surface-localised transporters as well as organellar transporters.

Thanks also for clarifying the methods used to generate the list of candidate transporters. This really improves the understanding of how this list was created.

The figures have been greatly updated along with the legends, and this had made a significant impact in improving the manuscript.

The localisation and updated figures have streamlined the narrative of the outcome of this screen. This data will be very useful to the wider parasitology field and is an impressive amount of work.

The comments from the other reviewers are also well addressed.

Reviewer #2 (Remarks to the Author):

All my points addressed - thank you.

Reviewer #3 (Remarks to the Author):

I have looked over the revised manuscript and the responses to all three reviewers' comments. I am satisfied that the authors have answered appropriately all questions, with one reservation below, and I support progressing with the revised manuscript.

I have one comment that I believe the authors should address, ideally experimentally but at least in their Discussion section. As requested, the authors have added a new experiment localizing two subunits of the V-ATPase that plays a critical role in amastigote viability. The images in Fig. 7E show a localization to a structure that they interpret as endosomal vesicles. The endosomes do emerge from the flagellar pocket membrane, so given the location they observe that suggestion is reasonable. However, without a marker, it is difficult to distinguish such vesicles from the flagellar pocket itself, which is indeed located close to the kinetoplast that can be seen in purple nearby the green fluorescence in the images. Hence, it is possible that V-ATPase is located, partially or completely, in the flagellar pocket membrane. In this case, a plausible explanation for its function could be that it pumps cytosolic protons into the flagellar pocket, thus maintaining internal pH homeostasis in the amastigote.

>The reviewer raises an interesting and important point. We report that V-ATPase signal is seen adjacent to the flagellar pocket and that the localization is consistent with endosomal compartments. Our interpretation is compatible with the reviewer's suggestion. There is dynamic turnover of membranes between the flagellar pocket and the endosomal system. A proportion of V-ATPase complexes may transit through the flagellar pocket at some stage. The experiments that would be required to conclusively demonstrate the direction of proton

pumping at that interface is far beyond the scope of the current manuscript. It is however a valid model that can be tested in future studies. We have now amended the text to take this into account (changes highlighted).

Line 367

“No signal was detected at the plasma membrane, suggesting the protective function of the V-ATPase under conditions of low external pH occurs at intracellular organelles and possibly at the flagellar pocket membrane.”

Line 467

“One simple model could be for the V-ATPase to extrude protons directly at the cell surface. Or data argues against a re-location of the V-ATPase to the cell surface, but one possibility that remains compatible with our localisation data is that cytosolic protons are extruded via the flagellar pocket. Or data also supports a function of the *L. mexicana* V-ATPase in organellar homeostasis: Both in promastigotes and amastigotes, the tagged *Leishmania* V-ATPase subunits (shown in Figure 7) consistently localised to internal foci, suggestive of endocytic organelles, and compatible with a fraction also localising to acidocalcisomes. This is consistent with other reports on the localisation of V-ATPase subunits in trypanosomatids, using proteomics methods [58, 59] or tagging and microscopy [60Billington, 2023 #1354, 61].”